# Polycomb enables primitive endoderm lineage priming in embryonic stem cells

Robert S Illingworth[1†], Jurriaan J Hölzenspies[2,3†], Fabian V Roske[2], Wendy A Bickmore[1*], Joshua M Brickman[2,3*]

[1]MRC Human Genetics Unit, Institute of Genetics and Molecular Medicine, University of Edinburgh, Edinburgh, United Kingdom; [2]The Danish Stem Cell Centre - DanStem, University of Copenhagen, Copenhagen, Denmark; [3]MRC Centre for Regenerative Medicine, Institute for Stem Cell Research, School of Biological Sciences, Univeristy of Edinburgh, Edinburgh, United Kingdom

**Abstract** Mouse embryonic stem cells (ESCs), like the blastocyst from which they are derived, contain precursors of the epiblast (Epi) and primitive endoderm (PrEn) lineages. While transient in vivo, these precursor populations readily interconvert in vitro. We show that altered transcription is the driver of these coordinated changes, known as lineage priming, in a process that exploits novel polycomb activities. We find that intragenic levels of the polycomb mark H3K27me3 anti-correlate with changes in transcription, irrespective of the gene's developmental trajectory or identity as a polycomb target. In contrast, promoter proximal H3K27me3 is markedly higher for PrEn priming genes. Consequently, depletion of this modification stimulates the degree to which ESCs are primed towards PrEn when challenged to differentiate, but has little effect on gene expression in self-renewing ESC culture. These observations link polycomb with dynamic changes in transcription and stalled lineage commitment, allowing cells to explore alternative choices prior to a definitive decision.

*For correspondence: Wendy. Bickmore@igmm.ed.ac.uk (WAB); Joshua.brickman@sund.ku.dk (JMB)

†These authors contributed equally to this work

Competing interests: The authors declare that no competing interests exist.

## Introduction

Embryonic stem cells (ESCs) derived from the ICM of the early mammalian blastocyst are characterised by their capacity to differentiate into all cell types of the future organism (pluripotency) and by their ability to transmit this property through successive self-renewing cell divisions. Pluripotency is supported by a combination of extrinsic signals - such as LIF - and by an intrinsic transcription factor (TF) network that includes the core pluripotency factors OCT4, SOX2 and NANOG.

Like the peri-implantation ICM from which they are derived, ESCs are heterogeneous and consist of at least two morphologically indistinguishable cell types, representing primed progenitors of the epiblast and endoderm lineages (*Singh et al., 2007*; *Canham et al., 2010*; *Lanner and Rossant, 2010*; *Morgani et al., 2013*). *In vivo* these progenitors exist very transiently prior to implantation, at which point cells rapidly become committed to adopt embryonic or extra-embryonic fates. However, in ESC culture, these two cell states dynamically interconvert and are maintained indefinitely (*Canham et al., 2010*). When isolated and challenged to differentiate, the PrEn-primed pluripotent population exhibits an enhanced capacity for endoderm differentiation in vitro and can colonise the extra-embryonic endoderm when re-introduced into either morulas or blastocysts (*Canham et al., 2010*; *Morgani et al., 2013*). The Epi-primed pluripotent population shows an equivalent enhanced capacity to differentiate towards epiblast lineages and contributes to the epiblast in vivo (*Canham et al., 2010*; *Morgani et al., 2013*).

At a molecular level, Epi-primed ESCs display elevated expression of mRNAs for various pluripotency associated TFs, such as *Nanog*, *Esrrb* and *Zfp42* (Rex1). In turn, the PrEn fraction expresses

higher levels of endoderm specific mRNAs (*Canham et al., 2010*; *Morgani et al., 2013*). Both populations express similar levels of *Pou5f1* (Oct4) and the ESC-specific cell surface markers SSEA1 and PECAM. In total, several hundred genes show small, but significant, changes in expression as ESCs transit between these primed states (*Canham et al., 2010*). How the expression of these genes is coordinately changed, and how this is linked to functional priming is unknown.

Polycomb and trithorax chromatin modifying complexes have been implicated in establishing the competence of ESCs to differentiate. Mouse embryos deficient for polycomb complexes PRC1 and PRC2 fail to develop beyond gastrulation and exhibit defects in both embryonic and extra-embryonic development (*Faust et al., 1995*; *Faust et al., 1998*; *O'Carroll et al., 2001*; *Voncken et al., 2003*). PRC mutant ESCs express high background levels of differentiation-specific determinants and are unable to down-regulate TFs associated with pluripotency during differentiation. Moreover, reprogramming of somatic cells to the pluripotent state (iPS cells) requires both PRC1 and PRC2 (*Pereira et al., 2010*).

PRCs orchestrate developmental programmes by maintaining target genes in a poised transcriptional state (*Dellino et al., 2004*; *Stock et al., 2007*). PRC2 trimethylates histone H3 at lysine 27 (H3K27me3) (*Cao et al., 2002*) through the EZH1/2 histone methyltransferase (HMTase) component of the complex and this histone modification can in turn recruit PRC1 through the chromodomains of CBX subunits (*Morey and Helin 2010*). Recently, variant PRC1 complexes have been shown to nucleate PRC2 binding providing a self-reinforcing mode of polycomb recruitment (*Blackledge et al., 2014*; *Cooper et al., 2014*). In mouse ESCs, H3K27me3 and PRCs occupy large domains at repressed genes that encode developmental regulators (*Boyer et al., 2006*; *Mikkelsen et al., 2007*; *Endoh et al., 2008*; *Ku et al., 2008*) and consequently transcripts of these genes are upregulated in response to loss of PRC1 or PRC2 in ESCs (*Boyer et al., 2006*; *Endoh et al., 2008*). The trithorax system is associated with trimethylation of histone H3 lysine 4 (H3K4me3) - a modification found at the majority of non-methylated CpG islands (CGIs) that marks a transcriptionally permissive state (*Klose et al., 2013*). A subset of CGIs, however, contain nucleosomes marked by both H3K27me3 and some H3K4me3, a combination of histone modifications referred to as bivalency (*Azuara et al., 2006*; *Bernstein et al., 2006*; *Mikkelsen et al., 2007*; *Voigt et al., 2012*; *Hu et al., 2013*; *Denissov et al., 2014*). The co-incidence of these two histone modifications is also accompanied by the presence of a form of RNA polymerase II associated with transcription initiation (*Brookes et al., 2012*), consistent with the hypothesis that the bivalent chromatin state contributes to robust gene activation or silencing during the exit from pluripotency and the initiation of differentiation (*Voigt et al., 2012*). Whether this chromatin state is involved in lineage priming or is required only for commitment during differentiation has not been explored.

PrEn- and Epi-primed ESCs can be isolated based on the expression of a highly sensitive fluorescent reporter that contains a Venus insertion in the primitive endoderm gene *Hhex*, and on the presence of the ESC-specific cell surface marker SSEA1 (*Canham et al., 2010*). Cells that express the *Hhex*-Venus (HV) reporter and SSEA1 self-renew, are PrEn-primed and readily convert to an HV$^-$, SSEA1$^+$, Epi-primed ESC state. Here we use this tool to investigate the role of polycomb in maintaining the balance between Epi- and PrEn-primed ESC populations. We use gene expression microarrays in combination with genome-wide nuclear run-on sequencing (GRO-seq) to identify a set of genes with a distinct expression profile in these populations that is primarily regulated at the level of transcription. We show that gene body H3K27me3 is reduced with increased transcriptional activity across our priming gene sets, suggesting that H3K27me3 is a response to these small and dynamic changes in transcription. In contrast, promoter proximal H3K27me3 and H3K4me3 levels are invariant during priming, with a pattern of H3K27me3 around the start site that is significantly enriched in PrEn priming genes. Consistent with this observation, single cell qPCR analysis on polycomb mutant ESCs suggests that PRC1 and 2 are required for the establishment of ESC heterogeneity. Furthermore, short-term inhibition of EZH2 activity results in reduced neural differentiation efficiency in vitro and preferential PrEn contribution in vivo. These results suggest that polycomb regulates rapid and reversible damping of transcription and the maintenance of heterogeneity in early lineage specification.

## Results

### Heterogeneous *Hhex* reporter expression reflects changes in transcription

To identify a definitive set of genes correlated with lineage priming in ESCs we isolated SSEA1+ (S$^+$) HV reporter ESCs by fluorescence activated cell sorting (FACS) based on the level of HV expression (low/high fractions; *Figure 1A*; green and red gates respectively). Amplified poly-A RNA from both populations was hybridised to expression microarrays and combined with data from a previous dataset (*Canham et al., 2010*). As previous analysis suggested that the PrEn-primed population (HV$^+$S$^+$) exhibited small but detectable increases in expression of endoderm markers, we first defined the set of genes showing differential expression between the undifferentiated Epi-primed fraction (HV$^-$S$^+$; *Figure 1A*; green gate) and spontaneously differentiated endoderm cells (HV$^+$S$^-$; *Figure 1A*; yellow gate) using analysis of variance (ANOVA; FDR of 0.05 and a fold change of > 1.5; *Figure 1B*). From this initial gene list, we then selected only those genes that showed a differential magnitude of expression between primed ESC populations, i.e. HV$^-$S$^+$ versus HV$^+$S$^+$ (*Figure 1A*; green and red gates respectively) consistent with the direction of differential expression in the original HV$^-$S$^+$ versus HV$^+$S$^-$ comparison (*Figure 1B*). Using these criteria we identified 210 genes that were up-regulated in HV$^-$S$^+$, and 533 genes up-regulated in the HV$^+$S$^+$ populations (*Figure 1B* and *Supplementary file 1*). These genes showed a subtle yet robust level of differential expression characteristic of these dynamic cell populations (*Figure 1C*). To validate these results, quantitative RT-PCR was performed on 2 independently grown clones of 2 independent *Hhex* reporter ESC lines. This confirmed the trajectory of differential expression for two randomly chosen panels of candidate genes with elevated expression in the Epi- and PrEn-primed populations, respectively. Moreover, the expression of *Nanog* but not *Oct4* was shown to be elevated in the Epi-primed population as has been previously reported (*Figure 1D*) (*Canham et al., 2010*).

Consistent with previous findings, HV$^+$S$^+$ cells exhibited elevated expression of genes characteristic of PrEn (*Sox7*, *Dab2*, *Vim* and *Furin*) (*Campe et al., 2013*; *Drexler et al., 2013),* while HV$^-$S$^+$ cells showed elevated expression of pluripotency or early epiblast markers, such as *Dppa3* (Stella), *Spic* and *Nle1* (*Supplementary file 1*) (*Dillner et al., 2013*; *Olsson et al., 2014*). Interestingly however, functional analysis showed that priming genes were not principally developmental in nature: Epi priming genes are involved in cell cycle and cellular metabolism, whereas PrEn priming genes are important for intracellular transport, cell migration and cell adhesion (*Supplementary file 2*). This suggests that cellular composition and metabolism is markedly altered, albeit reversibly, prior to the upregulation of developmental genes and the entry of these cells into lineage specification.

Physical characterisation of both sets of genes identified an elevated G+C composition and CpG observed/expected ratio surrounding their transcription start sites (TSSs; ± 500 bp; *Figure 1—figure supplement 1A*) consistent with a significant enrichment for CGI promoters (*Figure 1—figure supplement 1B*; p=2.7 × 10$^{-3}$ and 7.7 × 10$^{-11}$ for HV$^-$S$^+$ and HV$^+$S$^+$ populations respectively). This is of particular interest as CGIs provide a recruitment platform for a range of chromatin modifying activities, which can either promote or restrict transcription (*Klose et al., 2013*).

To determine if the differential mRNA levels observed between the HV$^+$S$^+$ and HV$^-$S$^+$ populations reflect changes at the level of transcription, we utilised a variation of GRO-seq to measure nascent RNA arising from engaged RNAP II in the primed cell populations (*Core et al., 2008*). Control profiles for *Actb* and the transcriptionally silent *Hoxd* cluster showed the expected high and low GRO-seq signals respectively (*Figure 1—figure supplement 1C*). Candidate priming genes *Six1*, *Wdr31*, *Stx2* and *Capn2* showed differential transcription that reflected the steady state mRNA analysis (*Figure 1E*). To determine if this was characteristic of priming genes, GRO-seq signal was mapped to a composite gene of normalised length. Genes upregulated in either the Epi-primed (HV$^-$S$^+$) or PrEn-primed (HV$^+$S$^+$) population displayed increased nascent transcript abundance in the relevant cell population (*Figure 1F,G*; *Figure 1—figure supplement 1D*) - albeit with a smaller magnitude of change than that observed for steady state RNA (*Figure 1—figure supplement 1E*). In addition, the average level of GRO-seq signal on PrEn priming genes was much lower than for Epi priming genes, while steady state mRNA levels were approximately equal (*Figure 1—figure supplement 1E*; boxplots at the bottom of the graph), suggesting that PrEn genes are at least in part regulated at a post-transcriptional level. These data indicate that gene expression patterns characteristic of priming

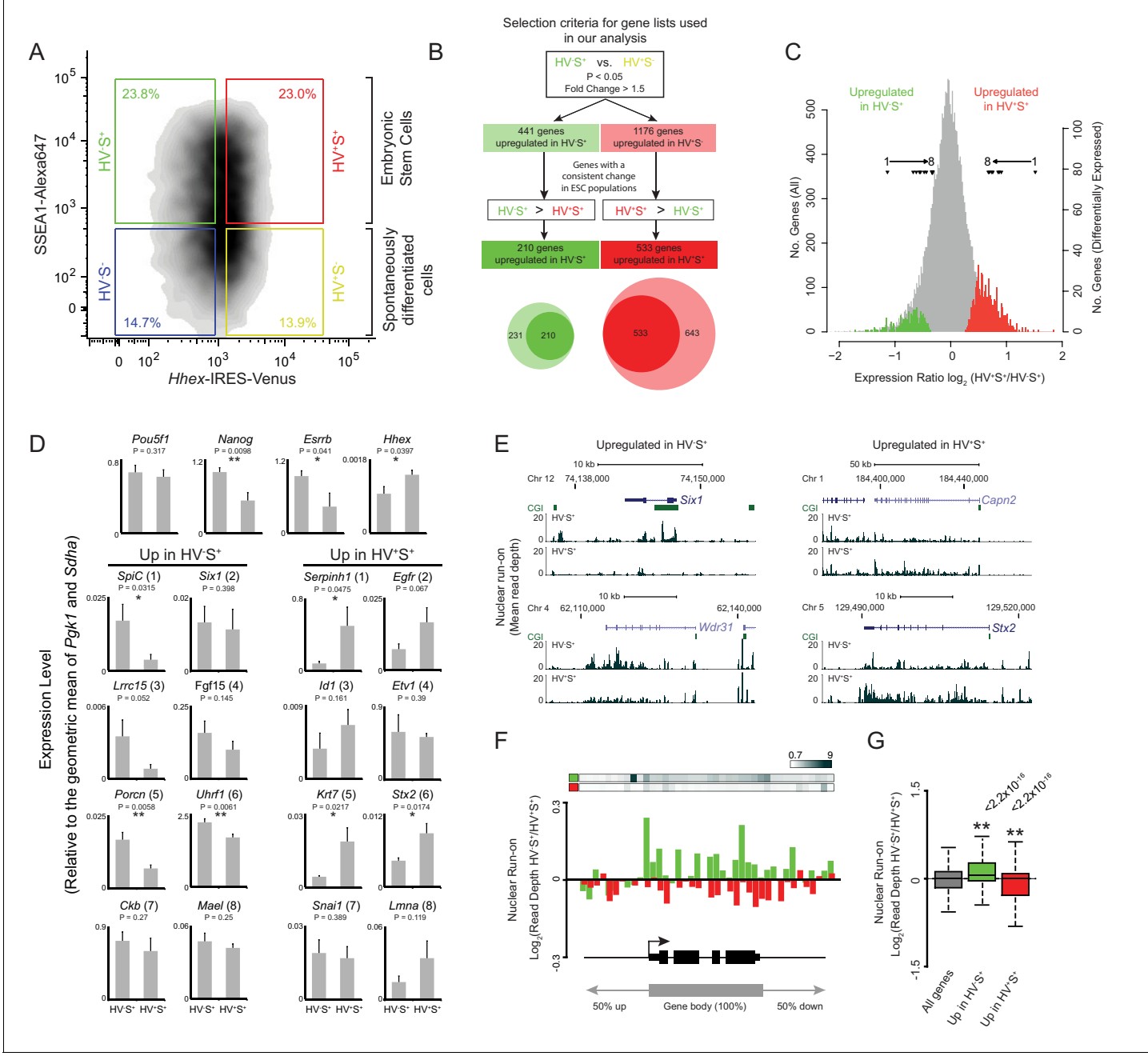

**Figure 1.** Heterogeneous *Hhex* reporter expression reflects coordinate changes in transcription. (**A**) FACS density plot showing ESC populations in *Hhex*-Venus reporter cells co-stained for the pluripotency marker SSEA1 by indirect immunofluorescence. Coloured boxes show the gating used to define the two spontaneously differentiated cell populations, HV⁻S⁻ (blue) and HV⁺S⁻ (yellow), and the two primed ESC populations, HV⁻S⁺ (green) and HV⁺S⁺ (red). (**B**) Strategy used to select gene lists for analysis. The Venn diagram shows the overlap between the genes identified by comparing HV⁻S⁺ to HV⁻S⁻ (light circles) and those with a consistent gene expression change when comparing HV⁻S⁺ to HV⁺S⁺ (dark circles). (**C**) Distribution of log₂ HV⁺S⁺/HV⁻S⁺ expression ratios for all genes (grey), genes upregulated in HV⁻S⁺ (green), and genes upregulated in HV⁺S⁺ (red; see also ***Figure 1—figure supplement 1***). The location of differentially expressed genes which are assessed by quantitative RT-PCR in (**D**) are indicated (filled triangles). (**D**) Quantitative RT-PCR validation of the microarray results. Bar graphs showing the mean + standard error of 4 biological replicates (2 independently grown clones of 2 independent *Hhex* reporter cell lines; *Hhex*-IRES-Venus (HV) and *Hhex*-3xFLAG-IRES-*H2b*-mCherry (HFHC)). The differential expression levels of these candidate genes are indicated in (**C**) and are ranked accordingly (labeled 1–8). Statistical analysis was performed using a homoscedastic one-tailed t-test (significant results are indicated* and p values are given above each graph). Samples were sorted by FACS as indicated in (**A**). For primer details see ***Supplementary file 5*** and for additional qPCR validation, see (***Canham et al., 2010***). (**E**) Example GRO-seq profiles (normalised read depth) at differentially expressed genes *Six1* and *Wdr31* (upregulated in HV⁻S⁺) and *Capn2* and *Stx2* (upregulated in HV⁺S⁺). Profiles

*Figure 1 continued on next page*

*Figure 1 continued*

were generated using the UCSC genome browser with bp coordinates given for mm9. (F) $Log_2$ GRO-seq HV⁻S⁺/HV⁺S⁺ read depth across a composite gene with normalised length representative of genes upregulated in HV⁻S⁺ (green) and genes upregulated in HV⁺S⁺ (red). The bars at the top show the average GRO-seq signal in each of the gene lists, while the graph shows the average differential transcript level for each of the gene lists. (G) Boxplots showing the log2 ratio of GRO-seq signal between HV⁻S⁺ and HV⁺S⁺ ESC populations for all genes (grey), genes upregulated in HV⁻S⁺ (green) and genes upregulated in HV⁺S⁺ (red). Two-sample permutation tests (oneway test) were used to compare the GRO-seq signal at priming vs all genes. p values of < 0.01 are indicated** and noted above the plot. See *Figure 1—figure supplement 1D* for individual biological replicates.

The following figure supplement is available for figure 1:

**Figure supplement 1.** Validation and characterisation of priming genes.

are primarily regulated at the level of transcription, but for PrEn priming transcripts there is also a post-transcriptional component.

## Priming genes are polycomb targets

To investigate mechanisms governing the dynamic transcriptional changes characteristic of lineage priming, we profiled histone modifications in primed populations of ESCs by native chromatin immunoprecipitation (ChIP) in two biologically independent cultures followed by deep sequencing. Given the proposed role of polycomb in the balance between pluripotency and differentiation (*Voigt et al., 2012*) and the enrichment of CGIs associated with priming gene sets (*Figure 1—figure supplement 1A,B*), we focussed on H3K27me3 and H3K4me3. As expected, the constitutively highly expressed *Actb* was devoid of H3K27me3 (*Figure 2A*) and enriched for H3K4me3 (*Figure 2—figure supplement 1A*). In contrast, a canonical polycomb target region, the *Hoxd* cluster, was decorated with H3K27me3 and had low level H3K4me3 in both HV⁺S⁺ and HV⁻S⁺ cell populations. *Zmiz1*, a gene upregulated in the HV⁺S⁺ population, showed a broad domain of H3K27me3 peaking at two CGIs, but the levels of H3K27me3 at *Zmiz1* were somewhat reduced in HV⁺S⁺ as compared to HV⁻S⁺ cells (*Figure 2A*). The reverse held true for genes upregulated in the HV⁻S⁺, Epi-primed population. For example, *Aldh1b1* - an HV⁻S⁺ enriched transcript - had slightly increased H3K27me3 that correlated with its reduced transcription in the HV⁺S⁺ fraction, despite a less extensive coating of H3K27me3 at this gene than at *Zmiz1*. *Zmiz1* and *Aldh1b1* had H3K4me3 surrounding their promoters, coincident with CGIs (*Figure 2—figure supplement 1A*; green bars), but unlike H3K27me3, their levels of H3K4me3 appeared not to vary between the primed cell populations. Since these differences were very subtle on single genes, we examined the average differential levels of H3K27me3 between the populations for all genes within each gene list to establish whether this change was specific to PrEn or Epi priming genes. We found no quantitative difference at the TSS (± 500 bp) between the HV⁻S⁺ and HV⁺S⁺ populations (*Figure 2B*). Meta-analysis confirmed the invariant H3K27me3 signal both at and upstream of the TSS, but surprisingly identified a small but significant reciprocal relationship between transcriptional activity and the level of H3K27me3 within gene bodies (TSS +501 bp to transcription end site (TES) +2 kb; *Figure 2B,C*). This occurred not only for PrEn priming genes, but also for Epi priming genes, despite the reduced magnitude and incidence of H3K27me3 TSS 'peaks' at these genes (*Figure 2B–E*). As the observed effect was subtle however, we replicated these results using an alternative, formaldehyde cross-linked ChIP methodology. This approach confirmed both the invariant H3K27me3 levels at the TSS of priming genes and the differential H3K27me3 within gene bodies (*Figure 2B*). While both ChIP methodologies showed very similar results, the gene-body effect at Epi-priming genes showed the same trend as in the native ChIP, but as the level of H3K27me3 is low in these genes and cross linked ChIP is less efficient than that performed under native conditions, this trend was less robust than that observed under native conditions.

The identification of genes with H3K27me3 enriched TSSs, coupled with the reciprocal relationship observed between gene body H3K27me3 and transcriptional activity, prompted us to investigate if these two phenomena were coincident on the same genes. However, when we split our gene lists based on the presence or absence of a 'peak' of H3K27me3 at the TSS, we found that the reciprocal relationship between H3K27me3 and transcription occurred irrespective of this sub-division (*Figure 2—figure supplement 2A,B*). Conversely, when we selected only those genes that showed

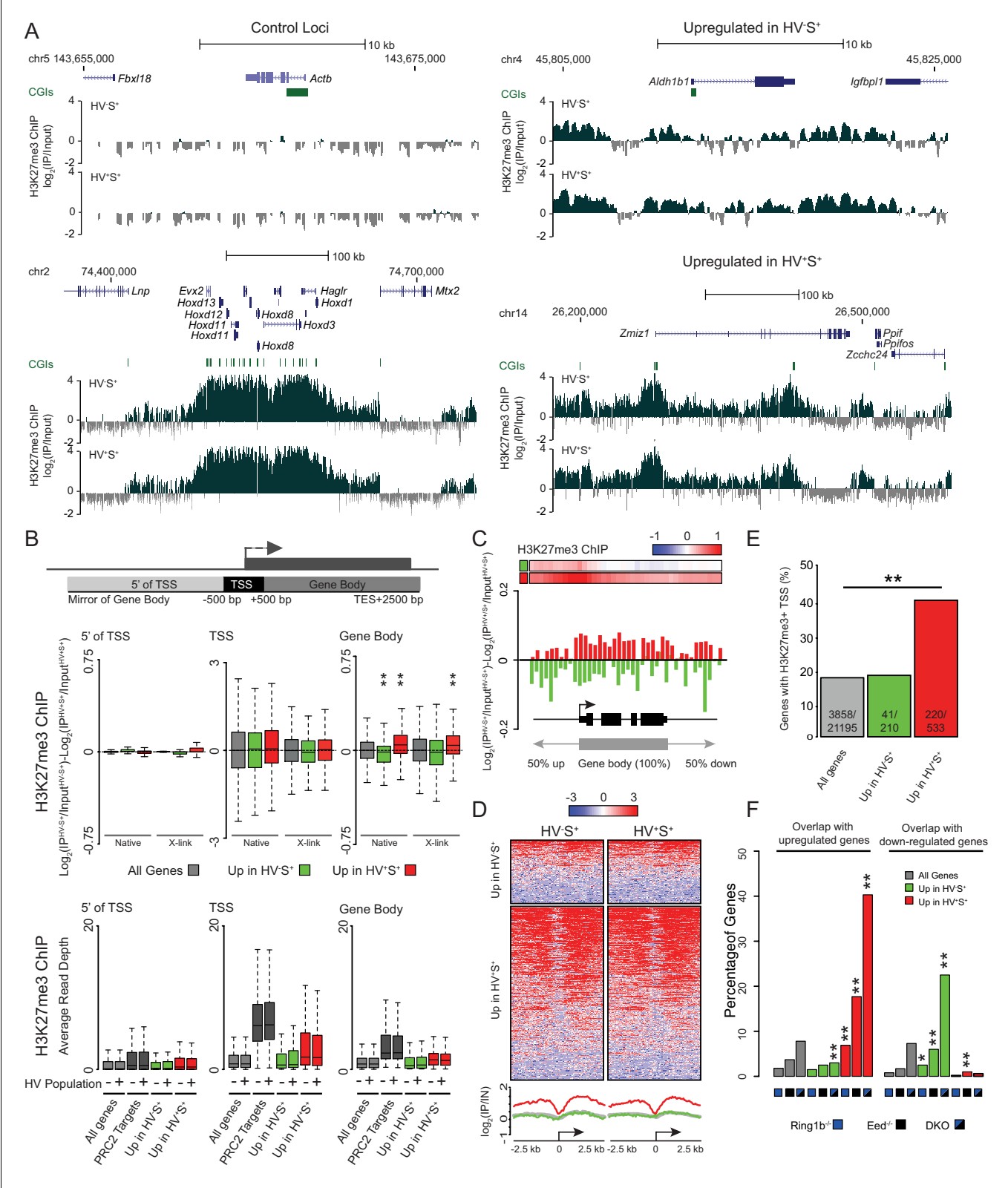

**Figure 2.** PrEn priming genes are enriched for polycomb targets. (**A**) Example H3K27me3 ChIP-seq profiles (log₂(IP/Input)) for *Actb* and *Hoxd* (Control Loci), *Aldh1b1* (Upregulated in HV⁻S⁺) and *Zmiz1* (Upregulated in HV⁺S⁺). Coordinates given are for the mm9 genome build. (**B**) Schematic representation of the stratification used to bin H3K27me3 signal with respect to genes (upper panel) and boxplots showing the ratio and average read depths (middle and lower panels respectively) of H3K27me3 ChIP-seq signal. The ratios represent the comparison of H3K27me3 ChIP-seq signal

*Figure 2 continued on next page*

*Figure 2 continued*

between the HV$^-$S$^+$ and HV$^+$S$^+$ ESC populations for both native and formaldehyde cross-linked ChIP (indicated as 'native' and 'X-link' respectively; the data is presented as in *Figure 1G*). Genes were defined as 'PRC2 Targets' if they were enriched for H3K27me3 at their TSS (± 100 bp) and were also upregulated in cells deficient for the PRC2 component EED (*Leeb et al., 2010*). Two-sample permutation tests (oneway test) were used to compare the H3K27me3 ChIP-seq signal at priming vs all genes. P values of < 0.01 are indicated** (all indicated p values are < 2.2 × 10$^{-16}$). (C) H3K27me3 ratios (log$_2$(IP$^{HV-S+}$/Input$^{HV-S+}$)-log$_2$(IP$^{HV+S+}$/Input$^{HV+S+}$)) between the HV$^-$S$^+$ and HV$^+$S$^+$ populations presented as for *Figure 1F*. (D) TSS (± 2.5 kb) heatmaps and summary plots of H3K27me3 signal (log$_2$(IP/Input)) for genes with differential expression between the HV$^-$S$^+$ and HV$^+$S$^+$ ESC populations. (E) Barplot representing the percentage of genes that are enriched for H3K27me3 within 100 bp of their TSS. Significance enrichment versus all genes was tested using a Fisher's test and ** indicates a p value of < 0.01 (actual value of 3.4 × 10$^{-36}$). (F) Barplots depicting the proportion of priming-genes that are up- or down-regulated in Ring1b$^{-/-}$, *Eed*$^{-/-}$ or Ring1b$^{-/-}$/*Eed*$^{-/-}$ double knockout ESCs (blue, black and blue/black labelled bars, respectively). Significant enrichment or depletion relative to all genes was assessed using a Fisher's exact test and significant enrichment or depletion for each comparison is indicated (*0.05 > p ≥ 0.01 and **p < 0.01). Significant p values indicated*/** are 7.5 × 10$^{-3}$, 3.51 × 10$^{-11}$, 1.08 × 10$^{-33}$, 7.49 × 10$^{-89}$, 2 × 10$^{-2}$, 2 × 10$^{-3}$, 1.34 × 10$^{-11}$ and 2.99 × 10$^{-13}$ (left to right).

The following figure supplements are available for figure 2:

**Figure supplement 1.** Invariant H3K4me3 levels at Epi- and PrEn-primed genes.

**Figure supplement 2.** Dynamic changes in gene-body H3K27me3 levels contrasts with a specific enrichment of H3K27me3 at the promoters of PrEn-primed genes.

**Figure supplement 3.** More pronounced changes in H3K27me3 levels are observed in differentiating cells.

**Figure supplement 4.** Differential gene-body H3K27me3 and the re-equilibration dynamics of *Hhex* defined lineage priming.

differential levels of GRO-seq signal between the primed populations (*Figure 2—figure supplement 2C*) only a subset of these genes showed an enrichment for H3K27me3 at their TSS (*Figure 2—figure supplement 2D*). These findings suggest that H3K27me3 depositions at the TSS and within the gene body are independent of one another. We used the same methodology to examine the behaviour of H3K4me3 in primed ESC populations. As expected, we found little variation in H3K4me3 levels either at the TSS or across gene bodies, for both sets of differentially expressed genes in both ESC populations (*Figure 2—figure supplement 1B,C*). Heatmaps representing the distribution of H3K4me3 around the TSSs (± 2.5 kb) of all differentially expressed genes showed that H3K4me3 levels did not change significantly between the HV$^-$S$^+$ and HV$^+$S$^+$ cell populations (*Figure 2—figure supplement 1D*) and generally appeared similar irrespective of the expression state in PrEn- and Epi-primed ESCs. In contrast, a far greater proportion of genes upregulated in the HV$^+$S$^+$ population showed promoter proximal enrichment for H3K27me3 (*Figure 2D*), and this corresponded to a significant enrichment for TSS associated polycomb 'peaks' at these genes (41.3% of genes upregulated in the HV$^+$S$^+$ population vs. only 19.5% of genes upregulated in the HV$^-$S$^+$ population; *Figure 2E*). Consistent with this finding, PrEn priming genes show an elevated tendency to be upregulated in response to the loss of both PRC1 and PRC2 components (RING1B and EED, respectively; *Figure 2F*). Conversely, Epi priming genes are enriched for genes that are downregulated in polycomb mutant ESCs (*Figure 2F*) (*Leeb et al., 2010*).

Given the relatively subtle nature of dynamic H3K27me3 redistribution within the body of priming genes we wanted to confirm this result using an independent dataset. Studies on ESC heterogeneity using the early epiblast marker *Rex1* (*Zfp42*) (*Marks et al., 2012*) included both gene expression and H3K27me3 ChIP-seq datasets. These populations dynamically interconvert, although unlike PrEn- and Epi-primed cells, the *Rex1* low (Rex$^-$) population contains partially differentiated cells and, therefore, Rex$^-$ cells give rise to *Rex1* high (Rex$^+$) cells at a lower frequency than the reverse. Thus, the Rex$^+$ population resembles the Epi-primed fraction, whereas the Rex$^-$ population includes spontaneously differentiated cells in addition to the PrEn-primed cells described here. Rex$^+$ and Rex$^-$ populations therefore resemble the HV$^-$S$^+$ and HV$^+$S$^+$ populations with the caveat that HV$^+$S$^+$ are merely a subset of Rex$^-$ cells. To explore this systematically we compared the gene set enriched in the *Rex1* expressing populations (*Supplementary file 3*) to the lists we compiled based on HV expression (*Figure 1*). Almost one third (73/210) of the Epi priming genes were also upregulated in the Rex$^+$ population (*Figure 2—figure supplement 3A*). Moreover, more than half of the PrEn

priming genes were represented in the list of genes upregulated in the Rex⁻ population (285/533 genes). Comparison of Rex⁺ with HV⁺S⁺ and Rex⁻ with HV⁻S⁺ showed negligible overlap (*Figure 2—figure supplement 3A*).

Consistent with our own findings in HV⁺S⁺/HV⁻S⁺ cells, differential expression was associated with a reciprocal enrichment of H3K27me3 across the gene bodies in Rex⁺/Rex⁻ cells (*Figure 2—figure supplement 3B*). In this comparison however, both differentially expressed gene sets show a significant reduction of H3K27me3 at the TSS as well as the gene body and this is not surprising as the Rex⁻ population contains cells that have already embarked on differentiation rather than just primed populations of ESCs. To test this contention, we performed ChIP-seq for H3K27me3 on spontaneously differentiated PrEn cells (HV⁺S⁻) and compared this to H3K27me3 levels in the Epi-primed population. Consistent with the Rex analysis, we observed more pronounced changes in H3K27me3 levels between these populations, including a significant reduction at the TSSs and upstream regions of PrEn-priming genes (*Figure 2—figure supplement 3C and D*).

These results confirm that PrEn priming genes are enriched for polycomb targets, but also suggest that changes in the levels of gene body H3K27me3 are inversely proportional to transcriptional changes in both primed and spontaneously differentiating ESCs, while measurable losses at TSSs only become evident upon differentiation.

In light of these findings, we investigated the relationship between the interconversion of the primed populations and differential gene-body H3K27me3 levels. To test this, we determined the rate at which primed populations re-equilibrate when returned to culture using modified reporters that contain either the original *Hhex*-IRES-Venus and a constitutive *H2b*-mCherry lineage label (HVHC ESCs) or an amplified *H2b*-mCherry under control of the *Hhex* locus and a constitutive *H2b*-Venus lineage marker (HFHCV ESCs). For both reporter lines, Epi- and PrEn-primed populations were isolated by FACS and returned to culture. *Hhex* expression was then monitored by flow cytometry at defined time points (18, 24, 48 and 72 h post-sorting; *Figure 2—figure supplement 4A,B*). Re-equilibration dynamics were found to be largely equivalent irrespective of the developmental trajectory or the reporter used. Moreover, co-culturing the two populations isolated from lineage labelled, but different colour reporter lines showed nearly identical dynamics to those cultured independently (*Figure 2—figure supplement 4B* – compare solid with dashed lines). These results indicate that the rate of interconversion was independent of the starting gene expression state and any paracrine interactions between the populations. By 72 h, both Epi- and PrEn starting populations were largely equivalent (*Figure 2—figure supplement 4A–C*), although complete equilibration was probably achieved around day 5 after sorting (extrapolated from the data in *Figure 2—figure supplement 4C*). Based on the re-equilibration dynamics observed in *Figure 2—figure supplement 4C*, we chose an intermediate time point during re-culture and determined the levels of gene body H3K27me3. Half of each of the primed populations used to generate the cross-linked ChIP data presented in *Figure 2B* was re-cultured for 30 h prior to performing H3K27me3 ChIP on the SSEA1+ (S⁺) fraction (*Figure 2—figure supplement 4D*). As expected (and comparable to *Figure 2A*), the ChIP-seq profiles for the *Hoxd* and *Actb* loci showed patterns of high and low H3K27me3 levels, respectively (*Figure 2—figure supplement 4E*). However, a comparison of the ChIP-seq signal between the re-cultured primed populations showed a complete loss of the differential H3K27me3 levels observed at the time of the initial sort (*Figure 2—figure supplement 4E* compared to the parental populations shown in *Figure 2B*). This suggests that differential gene-body H3K27me3 levels are closely coupled to the dynamics of the cells as they transit between primed states, supporting the idea that these patterns are dictated largely by gene transcription.

## Polycomb is required for transcriptional heterogeneity in mouse ESCs

PrEn priming genes are generally marked by invariant H3K27me3 proximal to their transcription start site, while epiblast priming genes are not (*Figure 2D,E*). Since both gene sets could be regulated by PRC2 in the gene body, we wanted to determine whether a complete removal of polycomb activity would result in a loss of priming. To determine whether polycomb functions as a primary regulator of coordinate changes in expression of priming-associated genes in individual ESCs we performed single cell qRT-PCR on SSEA1⁺ FACS sorted *Eed*⁻/⁻ (PRC2) and *Ring1b*⁻/⁻ (PRC1) ESCs and their wild type (WT) parental ESC line (assayed genes are shown in *Figure 3A,B* and *Figure 3—figure supplement 1*) (*Leeb et al., 2010*). Expression of housekeeping genes such as *Gapdh* and *Actb* and the core pluripotency factors *Sox2* and *Oct4* (*Pou5f1*) was homogeneous in both wild type and mutant

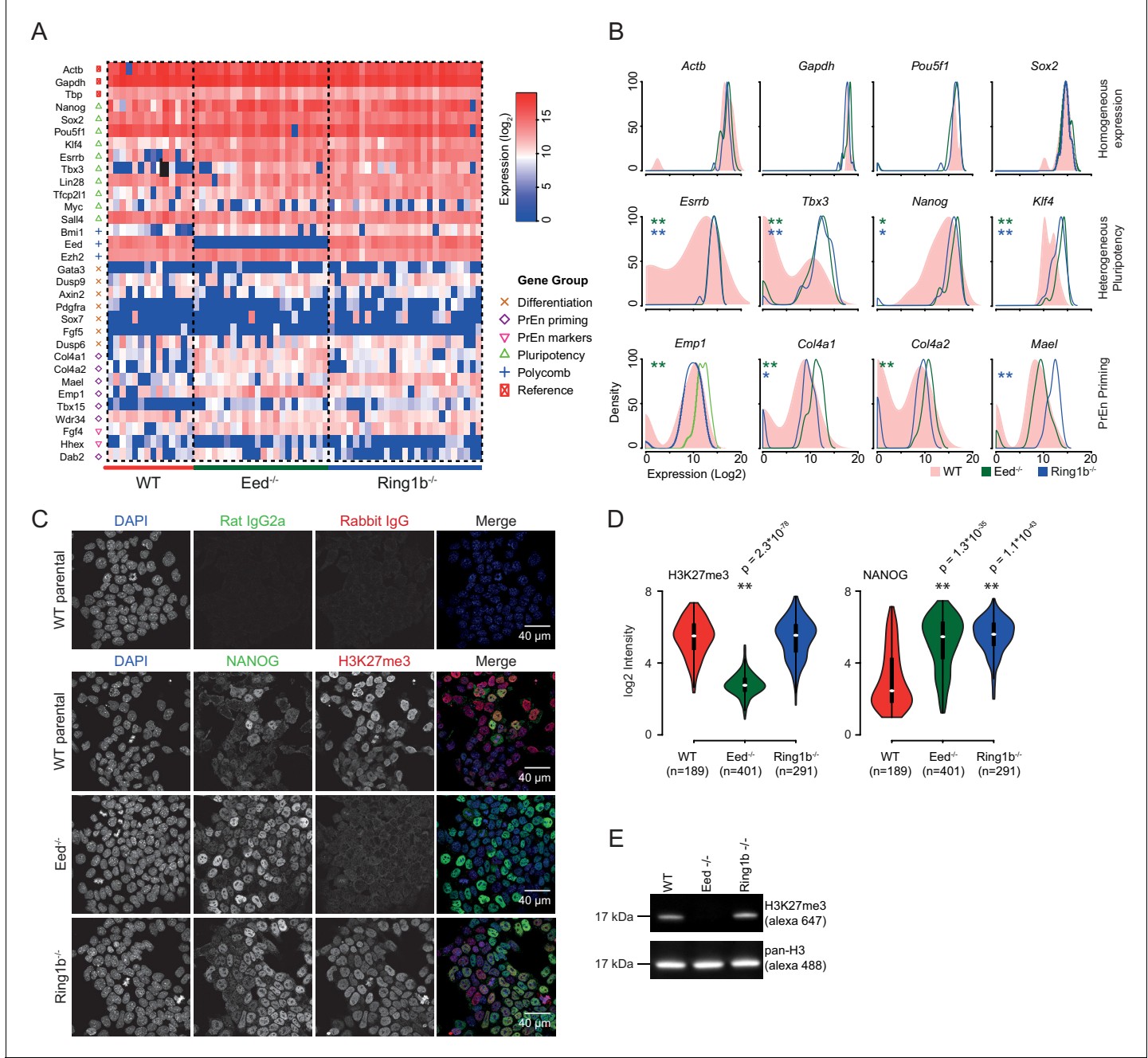

**Figure 3.** Polycomb deficiency results in reduced cell to cell expression heterogeneity of key pluripotency and PrEn priming genes. (A) Heatmap representing the log$_2$ expression values for individual cells for each gene (columns and rows respectively) measured by single cell qRT PCR (see also *Figure 3—figure supplement 1A*). Dashed lines, group cells based on genotype (WT - light red, *Eed*$^{-/-}$ - green and *Ring1b*$^{-/-}$ - blue). (B) Density plots showing the per cell distribution of log$_2$ gene expression values for each of the WT (light red), *Eed*$^{-/-}$ (green) and *Ring1b*$^{-/-}$ (blue) genotypes at a subset of key genes taken from (A). The expression profiles between WT and *Eed*$^{-/-}$ or WT and *Ring1b*$^{-/-}$ cells were assessed for significance using a Kolmogorov-Smirnov (KS) test under the null hypothesis that there was no shift in the distribution of expression values between these comparisons. Significant results are indicated (green and blue asterisks respectively; p values are presented in the left panel of *Figure 3—figure supplement 1B*). (C) Immunofluorescence labelling of NANOG (green) and H3K27me3 (red) in WT, *Eed*$^{-/-}$ and*Ring1b*$^{-/-}$ mouse ESCs. Scale bar = 40 μm. (D) Violin plots showing the quantification of immunofluorescent signal presented in (C). For each cell line, three fields of view were analysed and significance of the differential signal between WT and mutant cells was determined using a Wilcoxon Rank Sum test. Significant results are indicated with asterisks. (E) Example immunoblot showing H3K27me3 levels relative to total H3 in the indicated polycomb mutant ESC lines.

The following figure supplements are available for figure 3:

*Figure 3 continued*

**Figure supplement 1.** Single cell quantitative RT-PCR shows reduced cell to cell expression heterogeneity of key pluripotency and PrEn priming genes.

**Figure supplement 2.** *Eed* deficient ESCs cannot maintain *Hhex* priming in self-renewal.

cell lines. However, most PrEn priming genes and, surprisingly, several epiblast/pluripotency genes, were more homogeneously expressed in polycomb mutant cell lines compared to wild-type ESCs. Pluripotency genes such as *Nanog*, *Tbx3* and *Esrrb*, and PrEn priming genes such as *Col4a1*, *Col4a2*, *Mael* and *Emp1* all showed elevated and more homogeneous expression levels in the mutant vs. WT cells (*Figure 3A,B* and *Figure 3—figure supplement 1*). There were some exceptions, including PrEn markers *Hhex* and *Dab2*, which showed reduced average expression combined with reduced heterogeneity and *Wdr34* and *Fgf4*, which showed little to no change in their level or distribution of expression. Of the PrEn priming genes selected for single cell PCR, only one, *Tbx15*, showed an increase in heterogeneity when comparing mutant to WT cells (*Figure 3A* and *Figure 3—figure supplement 1A*).

Immunofluorescence (IF) for NANOG confirmed that the reduction in heterogeneity and increase in average expression level observed by single cell PCR was propagated to the protein level (*Figure 3C,D*). Both western blot and IF demonstrated the expected loss of H3K27me3 in the *Eed*[-/-] cells (*Figure 3C–E*). These results indicated that polycomb activity in ESCs is required to maintain the transcriptional heterogeneity that is associated with lineage priming. While Ring1b activity could possibly be compensated for by Ring1A (*de Napoles et al., 2004*; *Leeb and Wutz, 2007*; *Endoh et al., 2008*), the similarity in the phenotype between the *Eed* and Ring1b mutants suggests a general role for polycomb in ESC heterogeneity.

To better understand how priming is affected by the loss of polycomb activity, we employed the CRISPR system to generate an *Eed* deficient form of the HV5.1 reporter cells. Knockout of *Eed* was confirmed by Western blot for both EED protein and the associated H3K27me3 (*Figure 3—figure supplement 2A*). These HV5.1 *Eed*[-/-] ESCs showed a reduction in the cell surface pluripotency markers PECAM1 and SSEA1 when cultured in serum + LIF, but did not show spurious differentiation as assessed by morphology and could be maintained in serum + LIF without a loss of self-renewal. This reduction in pluripotency marker expression was partially rescued by addition of 3 μM Chiron to the medium (*Figure 3—figure supplement 2B*). Under these conditions, HV expression was also very modestly reduced, reflecting the same trend as observed in the single cell PCR (*Figure 3A*). However, addition of Chiron resulted in a significant expansion of a high *Hhex* expressing population. As *Hhex* is regulated by Wnt signaling (*Zorn et al., 1999*; *Huelsken et al., 2000*; *Rodriguez et al., 2001*; *Zamparini et al., 2006*) this increase in HV could reflect the removal of PRC2 activity that normally constrains high level *Hhex* expression in ESCs.

## Short-term inhibition of PRC2 function results in enhanced PrEn priming

The phenotype of polycomb mutants could either be a direct effect of polycomb on transcriptional heterogeneity or a result of the ESC gene regulatory network (GRN) being reset in response to alterations in the expression of immediate early polycomb targets and the selective pressure of ESC growth. To distinguish between these possibilities, we cultured ESCs in the presence of a reversible inhibitor of EZH2 methyltransferase activity - EPZ6438 (henceforth referred to as EPZ) (*Knutson et al., 2013*) for 24 h, followed by a combination of in vitro and in vivo assessment of their differentiation potential (*Figure 4A*).

Following short-term (24 h) EZH2 inhibition, immunoblotting confirmed a marked reduction in global H3K27me3 levels in five independently grown ESC lines (*Figure 4B*). These global changes in H3K27me3 levels were confirmed by native ChIP-qPCR at promoters and gene bodies of priming genes (*Figure 4—figure supplement 1A*). ESCs can be cultured in defined conditions with small molecule inhibitors of MEK-ERK (PD03) and GSK3 (CHIRON) that block differentiation, a culture condition known as 2i (*Ying et al., 2008*). As inhibition of MEK-ERK signaling has been shown to maintain ESCs by blocking PrEn priming and differentiation (*Hamilton and Brickman, 2014*), we tested its impact on H3K27me3 levels and found that both PD03 alone and 2i resulted in a global increase

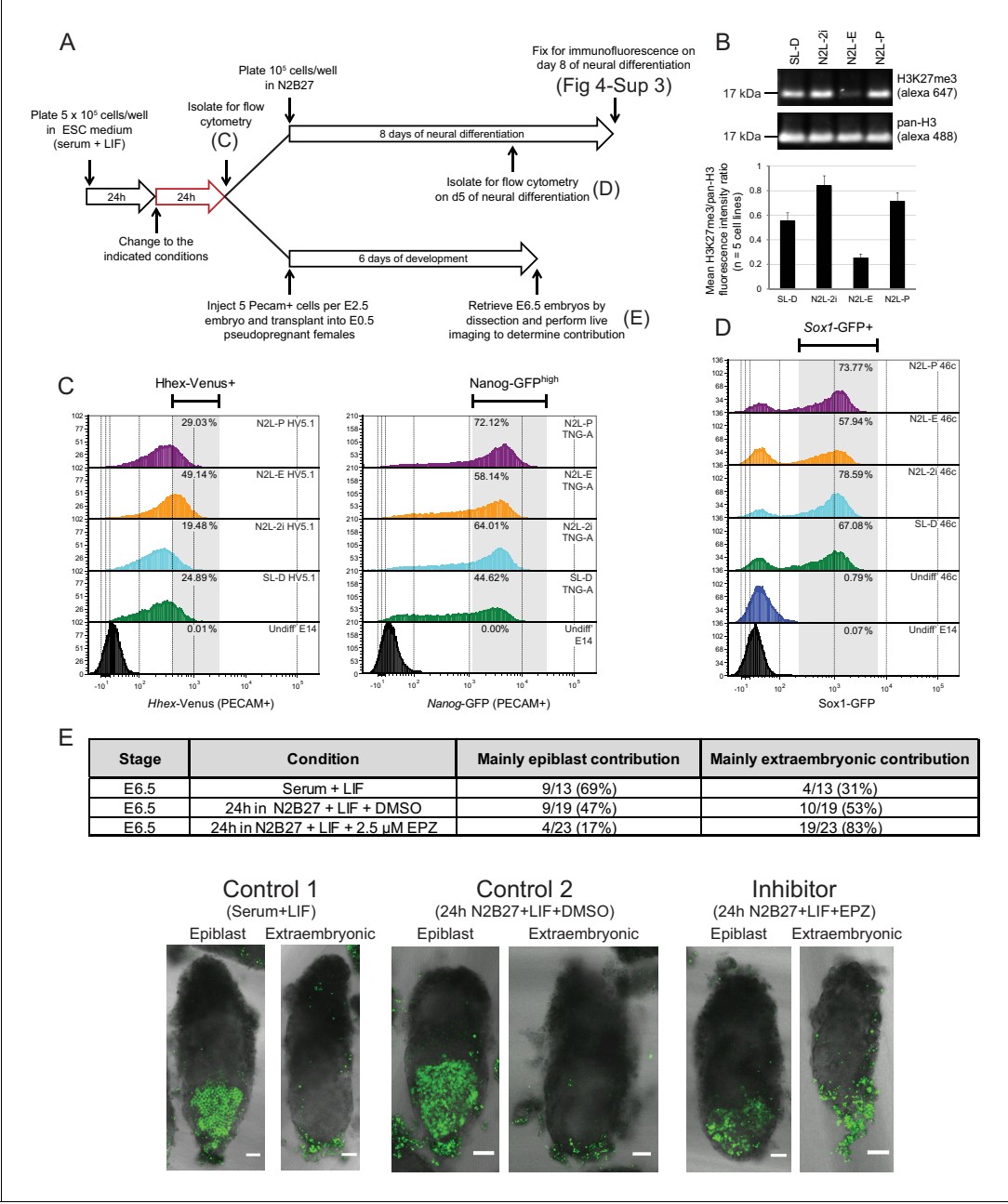

**Figure 4.** Blocking H3K27 methylation in ESCs promotes endoderm priming and the contribution to extra-embryonic endoderm in vivo. (**A**) Schematic overview of the experimental setup used to assess the in vitro and in vivo effects of EPZ on differentiation capacity. (**B**) Example immunoblot showing H3K27me3 levels relative to total H3 after 24 h of treatment in the indicated conditions. The densitometry ratio graph shows data from 5 different cell lines and error bars represent the SEM. (**C**) Flow cytometry after 24 h of treatment in the indicated conditions for the endoderm marker gene *Hhex* using HV5.1 ESCs (*Hhex*-IRES-Venus clone 5.1; (*Canham et al., 2010*)) and the epiblast marker Nanog using TNG-A ESCs (*Nanog*-GFP clone A; (*Chambers et al., 2007*)). Differentiating cells were excluded by gating for PECAM positive cells. The percentage of cells that fall within the indicated gate is shown in the upper right corner of each histogram. (**D**) Flow cytometry for the early neural marker Sox1 on 46c cells (*Sox1*-GFP; [*Ying et al., 2003*]) after 24 h of treatment in the indicated conditions followed by 5 days of neural differentiation. See also *Figure 4—figure supplement 3*. (**E**) Table showing the quantification of developmental potential, based on scoring E6.5 chimeric embryos as presenting with either 'mainly epiblast' or 'mainly extra-embryonic endoderm' contribution. Chi-square test result ($\chi^2$ = 9.96; p=0.007) shows a significant shift in the fraction of embryos with mainly extra-embryonic endoderm when comparing N2B27+LIF+EPZ (Inhibitor) with serum+LIF (Control 1) and N2B27+LIF+DMSO (Control 2). Representative examples of scored chimeric embryos are depicted for each condition. Treated ESC cells were FACs sorted for the cell-surface marker PECAM prior to injection into E2.5 embryos. (**C–E**) The conditions indicated are: N2L-P (N2B27 + LIF + 1 μM PD032); N2L-E (N2B27 + LIF + 2.5 μM EPZ6438); N2L-2i (N2B27 + LIF + 1 μM PD032 + 3 μM CHIR); SL-D (Serum + LIF + DMSO).

*Figure 4 continued on next page*

*Figure 4 continued*

The following figure supplements are available for figure 4:

**Figure supplement 1.** Reduction and redistribution of H3K27me3 in ESCs treated with EPZ and 2i, respectively.

**Figure supplement 2.** Global expression analysis of ESC following short-term EPZ treatment.

**Figure supplement 3.** A transient block of H3K27 methylation alters the efficiency of subsequent neuronal differentiation.

in H3K27me3 levels (*Figure 4B*). Re-analysis of published H3K27me3 distribution in serum + LIF versus 2i+LIF confirmed the previously observed loss of H3K27me3 at CGI promoter elements, alongside a reciprocal increase elsewhere, particularly within gene bodies (*Figure 4—figure supplement 1B–D*) (*Marks et al., 2012*; *Kumar et al., 2014*).

To directly test whether changes in H3K27me3 could alter lineage priming, we examined the global expression patterns in ESCs after 24 h of treatment under serum+LIF, N2B27+LIF or N2B27 +LIF+2i conditions, with or without EPZ (*Figure 4—figure supplement 2*). We observed very small sets of genes undergoing significant changes in gene expression, despite the clear and consistent global removal of H3K27me3 after 24 h of EPZ treatment (*Figure 4B*). Similar results were observed in different base media, although switching ESCs into different base media resulted in much larger changes in gene expression (*Figure 4—figure supplement 2A,B*; *Supplementary file 4*). Expression changes observed between conditions with and without EPZ occurred almost exclusively in those genes with H3K27me3 at their TSSs (*Figure 4—figure supplement 2C*), suggesting that the effects of EPZ on expression were directly related to the block of H3K27 methylation.

To understand the impact of this loss of H3K27me3 on the lineage priming status of ESCs, we assessed the effect of EPZ treatment on the fraction of ESC culture that is primed towards either PrEn or Epi as assayed by HV or Nanog-GFP transcriptional reporter ESC lines, respectively (*Figure 4C*). The response of these reporters to 24 h incubation in N2B27+LIF base media containing standard 2i or PD03 alone, conditions in which PrEn priming is specifically blocked by inhibiting MEK-ERK signaling (*Hamilton and Brickman 2014*), was compared to EPZ treatment in the same base medium, a condition in which H3K27me3 catalysis is blocked. We found that this short-term inhibition of H3K27me3 resulted in increased PrEn priming (HV$^+$ fraction in *Figure 4C*, left panel), although there is little specific effect of EPZ on Nanog expression (*Figure 4C*, right panel).

Lineage priming in ESCs has a clear functional definition that is based on differentiation outcome (*Canham et al., 2010*; *Morgani et al., 2013*) and we found that transient inhibition of H3K27me3 inhibited priming toward Epi derivatives and stimulated priming toward PrEn. To assess whether inhibition of H3K27me3 deposition influences ESC differentiation bias, we tested their tendency to differentiate in a specific direction upon removal of the reversible pharmacological block to H3K27 methylation. Epi differentiation was assessed in the form of defined neural differentiation using a reporter cell line in which GFP expression is driven by the promoter of the neural progenitor marker, Sox1. Following EPZ treatment (*Figure 4A*) Sox1-GFP (46c) cells were differentiated and the levels of GFP expression were quantified by flow cytometry on day 5 (*Figure 4D*). Transient removal of H3K27me3 resulted in reduced neural differentiation efficiency and cells that formed less extensive axon networks while expressing relatively high levels of TUJ1 (*Figure 4D*; *Figure 4—figure supplement 3*). We also assessed the capacity of the descendants of EPZ-treated ESCs to colonize the epiblast or PrEn lineages in vivo by injecting treated ESCs that contained a constitutive nuclear mCherry label (HVHC ESCs, see the methods section) into host morulas. Prior to injection, HVHC ESCs were grown in serum+LIF for 24 h, followed by 24 h of treatment in fresh serum+LIF (ESC control), N2B27+LIF+DMSO (DMSO control), or N2B27+LIF+EPZ (EPZ). ESCs grown in serum+LIF gave the expected high levels of Epi contribution and minimal contribution to the extra-embryonic visceral and parietal endoderm as we have observed in morula injections previously (*Morgani et al., 2013*). While ESCs cultured in neutral media (N2B27+LIF+DMSO) produced a modest enhancement in extra-embryonic contribution, they also gave robust contribution to the epiblast (control 2, *Figure 4E*). In contrast, descendants of EPZ-treated ESCs were found at low levels in the embryonic epiblast and throughout the extra-embryonic endoderm (*Figure 4E*). The impact of H3K27me3

inhibition on colonization of the PrEn and Epi lineages was apparent in both the number of embryos with extra-embryonic contribution and the extent of contribution in individual embryos (*Figure 4E* and data not shown). These results indicate that short-term inhibition of polycomb activity strongly reduces contribution to the epiblast while enhancing contribution to extra-embryonic endoderm. Taken together, both the in vitro differentiation and in vivo contribution results indicate that a transient loss of PRC2 catalytic activity shifts the distribution of priming in ESCs in favor of PrEn and away from epiblast.

## Discussion

Lineage primed cells, although functionally distinct, differ only subtly at the level of gene expression. These small fluctuations are associated with the acquisition of a distinct differentiation bias, although the corresponding changes in the physical properties of chromatin at individual gene loci are expected to be modest at best. This presupposes an inherent limitation in the ability to robustly detect the molecular events surrounding lineage priming on a per-gene basis. However, here we have taken advantage of the fact that these small expression changes occur synchronously across an extended set of genes. This allowed us to perform multi-gene meta-analysis for both transcriptional and epigenetic states and, in so doing, to detect changes that would otherwise have been unappreciated. This added power has allowed us to substantiate the gene expression changes observed previously (*Canham et al., 2010*; *Morgani et al., 2013*). Moreover it has allowed us to determine that priming in embryonic stem cell populations is principally regulated at the level of transcription and involves the activity of polycomb repressive complexes. While the polycomb system has long been associated with regulating developmental gene expression, here we show that these proteins are both enablers and regulators of priming toward specific lineages during sequential rounds of ESC self-renewal.

### Regulation of gene expression in ESC lineage priming

Lineage priming has been observed in several systems and refers to the existence of stem cell populations that possess equivalent capacities for self-renewal combined with transiently biased developmental potential (*Hu et al., 1997*; *Mansson et al., 2007*; *Kobayashi et al., 2009*; *Canham et al., 2010*; *Tsakiridis et al., 2014*). Some insight has been gained as to the signalling cascades that influence lineage priming in ESCs (*Hamilton and Brickman, 2014*), yet the gene regulatory mechanisms underpinning these subtle and transient expression changes have remained undetermined. Here we performed nascent RNA analysis on Hhex-defined Epi- and PrEn-primed ESC populations to show that the regulation of lineage priming occurs primarily at the level of transcription. A small number of genes do however show alterations in steady-state but not nascent RNA levels, suggesting that post transcriptional mechanisms also play a role, particularly for genes that are upregulated in the PrEn-primed population. These genes include immediate early responders to ERK/MAPK signalling, such as Fos, Jun, Fosl1, Fosl2, Ctgf, Cyr61 and Igf2, all of which were previously shown to be regulated at the level of RNA degradation (*Rabani et al., 2011*). As short durations of MEK-ERK signalling can stimulate the expression of these genes and promote reversible PrEn priming, a post-transcriptional mechanism would be compatible with rapid signal-induced state switching.

### Polycomb allows for dynamic modulation of the expression of both PrEn and Epi priming genes

The Polycomb repressive system underpins the regulation of developmental gene expression programs during embryogenesis (*Fisher and Fisher 2011*) and is important in lineage commitment (*Chamberlain et al., 2008*). We show that transcriptional changes associated with lineage priming in ESC populations correlate with levels of H3K27me3 in two ways. An invariant pattern of H3K27me3 proximal to the transcription start site of PrEn priming genes and a reciprocal relationship between transcription and the levels of gene body H3K27me3 (*Figures 1* and *2*). The dynamic changes in PRC2 activity within gene bodies links H3K27me3 deposition to changes in nascent transcription. This result suggests that PRC2 occupancy may be antagonised by local polymerase density and associated changes in chromatin state (*Schmitges et al., 2011*; *Miyazaki et al., 2013*). Thus our observations argue that polycomb is not merely a silencing factor, but can help balance the dynamic oscillations in transcription that occur during lineage priming. This general role for polycomb

contrasts with the lineage specific implications of the invariant pattern of H3K27me3 that we observe around the transcription start site of a large proportion of PrEn priming genes. Based on ChIP-seq analysis on mixed ESC populations, it has been suggested that polycomb targets in ESCs are lineage specific developmental genes, including PrEn genes (*Azuara et al., 2006*; *Bernstein et al., 2006*; *Mikkelsen et al., 2007*; *Fisher and Fisher, 2011*). However, these datasets contain a large number of genes expressed at unrelated stages of development. Here we link Polycomb specifically to the choice between Epi and PrEn differentiation that occurs in the blastocyst at around the time of ESC derivation (*Chazaud et al., 2006*). Based on our analysis, a large proportion of PrEn priming genes (~40%; *Figure 2E*) are polycomb targets that are upregulated in response to loss of polycomb activity (*Leeb et al., 2010*). However, datasets acquired using polycomb mutant ESCs are complicated by the selective nature of ESC growth and the capacity of polycomb mutant ESCs to adapt to culture over time. We also observed enhanced homogeneous expression of both PrEn genes and heterogeneously expressed members of the core Epi/pluripotency network (e.g. *Esrrb*, *Nanog* and *Tbx3*; *Figure 3*). One explanation for this lies in the inherently selective nature of ESC expansion. Thus, the induction of higher levels of PrEn priming might incur a growth disadvantage that could be compensated for by increased epiblast/pluripotent gene expression in PrEn primed cells. As PrEn supports Epi expansion in vivo (*Morgani and Brickman 2015*), a loss of polycomb might result in the production of paracrine factors that can promote epiblast/pluripotent gene expression as has recently been suggested for the Wnt/Pcp pathway downstream of Jarid2 (*Landeira et al., 2015*). In light of these results, we propose a model whereby polycomb acts around the start site of transcription to constrain PrEn gene expression within a limited dynamic range under homeostatic conditions, and in so doing, facilitates PrEn priming through transient and coordinated fluctuations in gene expression (*Figure 5A*). Sustained loss of polycomb activity therefore leads to selection of cells that exhibit reduced levels of heterogeneity.

Removal of H3K27me3 by short-term inhibition of polycomb (EZH2) methylation activity produces a state in which cells exhibit enhanced PrEn priming with very few actual changes in gene expression. This modest transcriptional effect could be due to the passive nature of H3K27me3 loss, where progressive dilution of the modified histone occurs as hypomethylated H3 is deposited following rounds of DNA replication (*Alabert and Groth, 2012*). Alternatively, removal of certain histone modifications may simply have little impact on global transcription under steady-state culture conditions, as the TF circuitry required to maintain gene expression levels remains intact. There is some evidence for this scenario, as disruption of components of both the SET1/MLL and PRC2 complexes leads to a marked reduction in the levels of H3K4me3 and H3K27me3, respectively, with surprisingly little effect on gene expression. Interestingly, in these instances, the limited effect on expression contrasts with a marked impact on differentiation potential and these findings are compatible with the observations presented here (*Carlone et al., 2005*; *Jiang et al., 2011*; *Clouaire et al., 2012*; *Riising et al., 2014*). We propose then, that EPZ treated cells are, to an extent, freed from the constraints on gene expression variability that are normally supported by polycomb. Under these conditions, priming loci have a greater probability of engaging in expression when challenged to differentiate. As a result, exposure of ESCs to short-term polycomb inhibition does not lead to a clear transcriptional response, but predisposes PrEn priming genes for rapid upregulation upon exposure to differentiation cues (*Figure 5B*). This is also consistent with the data we obtained using the *Eed* mutant HV reporter cell line, which showed enhanced expression of HV in response to Chi, a Gsk3 inhibitor also used to induce endoderm differentiation from ESCs. Taken together, our work suggests that stable polycomb occupancy at TSSs marks priming genes in ESCs and is coupled to highly dynamic, but limited fluctuations in transcription. These fluctuations are a component of the priming process, but their magnitude is kept in check to prevent ESCs from embarking on differentiation.

Embryonic development normally progresses from a state in which blastomeres co-express embryonic and extra-embryonic determinants, such as *Nanog* and *Gata6*, to states in which these factors are expressed in a mutually exclusive pattern in the ICM cells of the blastocyst in progenitors of the embryonic epiblast and extra-embryonic endoderm, respectively. The segregation of these two lineages is dependent on Fgf signalling (*Yamanaka et al., 2010*) and involves the induction of lineage specific transcription factors. ESCs are derived from these stages of embryonic development and our data suggests that polycomb group proteins are important mediators of these initial lineage segregation events, perhaps supporting a limited period of dynamic cell fate choice in PrEn

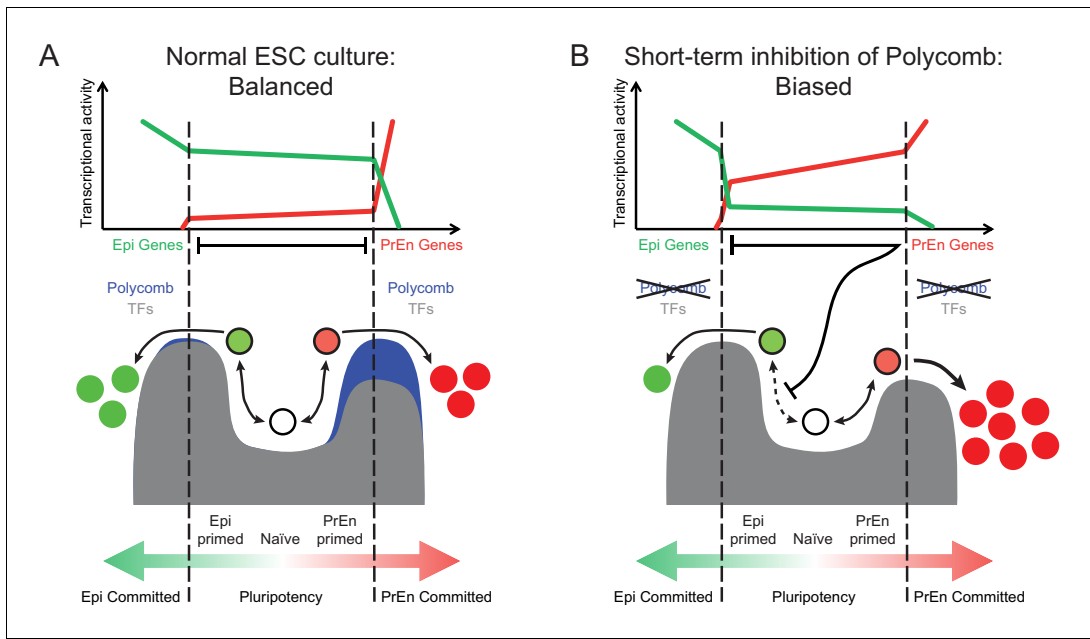

**Figure 5.** Model for the role of polycomb in lineage priming of ESCs. (**A**) The model depicts the stable interconversion of Epi- and PrEn-primed cell populations when ESCs are maintained in serum + LIF culture conditions. Epi- and PrEn-primed populations are represented as green and red filled circles, respectively. While these cells express differential levels of priming genes, they maintain ESC surface markers (represented by the black boundaries of the circles). Spontaneous differentiation in ESC culture is represented by a switch from ESC surface markers to lineage specific surface markers (represented by the coloured boundaries of the circles) as well as a switch from expression of priming genes to lineage specific marker genes (represented by the change from light to dark fill of the circles). Within the dashed vertical lines, the equilibrium between these states and self-renewal continues indefinitely. This balance is maintained, in part, by forces that resist lineage commitment to either the Epi or PrEn lineages (depicted as peaks). Both transcription factors (TFs; dark grey) and polycomb mediated repression (blue) are asymmetric components of this system, with TFs mainly restricting commitment towards Epi and polycomb mainly restricting commitment towards PrEn. (**B**) In the absence of polycomb-mediated repression, the balance is lost and the equilibrium shifts in favour of PrEn gene expression, which disrupts Epi priming and results in reduced Epi and increased PrEn commitment upon differentiation. Relative transcription levels of genes characteristic of Epi (green line) and PrEn (red line) are shown in the top panel.

progenitors that precedes commitment to embryonic or extra-embryonic lineages (*Xenopoulos et al., 2015*).

## Materials and methods

### ESC culture, inhibitor treatment and differentiation

E14 Tg2a ESCs (hereafter referred to as E14; 129/Ola background) and the derivative line HV5.1 (*Canham et al., 2010*) were cultured on 0.1% (w/v) gelatin (Sigma G1890) coated Corning flasks in ESC medium + LIF (GMEM (Sigma G5154) supplemented with 10% (v/v) fetal calf serum (FCS; Life Technologies, Carlsbad, CA 10270–106), 1x MEM non-essential amino acids (Life Technologies 11140–036), 0.1 µM 2-Mercaptoethanol (Sigma, St. Louis, MO M6250), 2 mM L-Glutamine (Life Technologies 25030–024), 1 mM Sodium Pyruvate (Life Technologies 11360–039), and 1000 U/ml of LIF. For passaging, 60–90% confluent culture flasks were washed with PBS, incubated for 2–3 min at 37°C in attenuated trypsin (0.025% (v/v) trypsin (Life Technologies 15090–046), 1.27 mM EDTA, and 1% (v/v) Chicken serum (Sigma C5405) in PBS (Sigma D8537)), and tapped to release ESCs from the flasks. Attenuated trypsin was inactivated by adding 9 volumes of ESC medium and this mixture was repeatedly pipetted to obtain a single cell suspension. ESCs were centrifuged, resuspended in ESC medium and replated onto gelatin coated flasks at a ratio of between 1:2 and 1:10, or at ~5 × 10$^4$

cells/cm$^2$ (determined using a hemocytometer - Neubauer). All centrifugation steps with live cells were performed at 330 x g for 3 min at room temperature (RT). For short term inhibition experiments (*Figure 4*), cells were plated in standard ES medium at $5 \times 10^4$ cells/cm$^2$ and cultured for 24 hr. The medium was then changed to defined medium (N2B27 + LIF) containing the following inhibitors individually or in combination: PD032 (1 μM), CHIRON (3 μM) and EPZ6438 (2.5 μM). For control conditions, the medium was changed to either standard ESC medium containing DMSO at the same concentration as treated conditions (primed control), or N2B27 + LIF + 2i (naïve control; (*Ying et al., 2008*)). For neural differentiation, cells were plated in N2B27 (200 μl/cm$^2$) and every day the medium was changed gently to avoid disturbing the cells (see also the scheme in *Figure 4A*).

All ESC lines used in this study were routinely karyotyped and tested for mycoplasma. All lines were found to be clear of mycoplasma and if lines developed an abnormal karyotype they were discarded. The Danish Stem Cell Center operates a mycoplasma free cell culture facility, any outside or primary lines being brought in must be screened in quarantine, before they can be imported into the facility.

## Generation of ESC lines

*Hhex*-3xFLAG-IRES-*H2b*-mCherry (HFHC) ESCs were generated from germ-line competent 129/Ola E14 ESCs (E14Ju) (*Hamilton and Brickman, 2014*) using a modified version of the plasmid used to create HV cells (*Canham et al., 2010*), pT1-HIV. In this modified version of the plasmid, pT1-HFHC, a triple FLAG-tag followed by a stop codon was inserted after the *Hhex* cDNA and the coding sequence for Venus was swapped out for *H2b*-mCherry. After electroporation, cells were expanded in serum + LIF under 150 μg/ml Hygromycin selection and correctly targeted clones were selected by Southern blot. The CMV-HyTK selection cassette was excised by lipofecting correctly targeted clones with a pCAG-Cre vector and two clones, HFHC10.4 and HFHC12.3, were selected to be used in our experiments based on their comparable *Hhex* reporter flow cytometry profiles and their behaviour in in vitro differentiation assays.

HVHC and HFHCV ESCs were generated from HV5.1 and HFHC10.4 ESCs, respectively, using pCAG-*H2b*-mCherry and pCAG-*H2b*-Venus vectors, respectively (a kind gift from Heiko Lickert, Helmholtz Zentrum München). After electroporation and 10 days of clonal expansion in serum + LIF with 1 μg/ml Puromycin, individual ESC colonies were picked, expanded and analysed by flow cytometry to select clones with a strong and constitutive mCherry (HVHC) or Venus (HFHCV) fluorescence, while having the same *Hhex* reporter fluorescence distribution as the parental cell line. All electroporations were performed at 250 V and 500 μF with 25 μg of linearised plasmid on 5 million cells of the parental cell line cultured in serum + LIF.

Eed$^{-/-}$ clones were generated by transfecting (Lipofectamine 2000, Life Technologies) HV5.1 ESCs with pX330 containing an *Eed*-specific guide sequence with a mismatched 5' guanine (GTTTTCGTC TCCCGAGAGGTC) as described (*Ran et al., 2013*). Further details of the CRISPR strategy and information on the *Eed*$^{-/-}$ and rescue ESCs used as Western blot controls are available on request.

## Antibody staining, flow cytometry and cell sorting

For cell surface marker analysis by flow cytometry, harvested ESCs were resuspended to single cell in SSEA1 Mouse anti-Mouse IgM-kappa primary antibody (clone MC480; Developmental Studies Hybridoma Bank; 1:1500 in FC buffer - 10% FCS in PBS) and incubated on ice for 15 min. After two washes in ice cold FC buffer, cells were incubated in Alexa Fluor 647-conjugated Goat anti-Mouse secondary antibody (Molecular Probes, Eugene, OR A21238; 1:1000 in FC buffer) on ice for 15 min, washed twice in FC buffer, and resuspended in 1 μg/ml DAPI in FC buffer to label dead cells. Alternatively, cells were directly labelled with either Mouse anti-Mouse Alexa Fluor 647-conjugated SSEA1 (1:400; BD Pharmingen, San Diego, DA 560120), or Rat anti-Mouse APC-conjugated CD31 (PECAM1; 0.5 μg/ml; BD Pharmingen 551262) and the ESCs processed as above. Forward and side scatter were used to filter out debris and cell clumps and DAPI staining was used to remove dead cells. Cells were run through a cell strainer (Corning, Corning, NY 352235) prior to sorting on a BD FACSAriaIII. Sorted cells were collected at 4°C in 15 ml falcon tubes (Corning 352054) containing ~0.2x the predicted final volume of FC buffer to ensure cells were collected in PBS with a final concentration of ~2% FCS. Different populations used in the same experiment were collected simultaneously and when multiple tubes of sorted cells were required, full tubes were stored on ice and

pooled at the end of the sorting session. Sorted cells were checked for sufficient purity (>80% of cells detected as being in the original sorting gate) by running ~1000 cells through the sorter again. Flow cytometry stacked histograms were generated using FCS Express 4 Flow Research (v. 4.07.0011).

## Immunofluorescence

ESCs were seeded at ~5 × 10$^4$ cells/cm$^2$ in standard ESC medium in 8-well microscopy slides (Ibidi; 80826) or 6-well culture plates (Corning) coated with 0.1% (w/v) gelatin. After two days, cells were fixed in 4% (w/v) paraformaldehyde diluted in PBS from 16% methanol-free formaldehyde stock (Pierce Biotechnologies, Waltham, MA; 28906) for 15 min at RT. The fixation was inactivated by adding 1.25 M glycine to a final concentration of 125 mM for 5 min at RT. Fixed cells were then washed briefly in PBS, permeabilised in PBS containing 0.3% (v/v) Triton X-100 (PBST) for 10 min at RT and blocked in 2% (v/v) Donkey serum in PBST for >2 hr at RT. Primary and secondary antibody incubations were done in PBST containing 1% (w/v) BSA either overnight at 4°C or for >2 hr at RT, and all antibody dilutions were cleared of aggregates prior to use by centrifugation at >17,000 x g for 15 min at 4°C. Antibody incubations were followed by 3 washes in PBST for 10 min at RT. After the final incubation, cells were washed twice in PBST for 10 min at RT and then stored in PBST containing 1 µg/ml DAPI at 4°C in the dark, packed in Parafilm to prevent evaporation. Antibodies were Rat anti-NANOG (2.5 µg/ml; eBioscience, San Diego, CA 14–5761), Rabbit anti-H3K27me3 (2 µg/ml; Millipore, Billerica, MA 07–449), Mouse anti-TUJ1 (2 µg/ml; Covance, Princeton, NJ mms-435p), Rabbit anti-GATA6 XP (1:1600 (~0.25 µg/ml); Cell Signaling, Danvers, MA 5851), Mouse anti-AFP (2.5 µg/ml; R&D Systems, Minneapolis, MN MAB1368) (Rat anti-RFP (1 µg/ml; Chromotek, Germany 5F8), Alexa Fluor 488-conjugated Rabbit anti-GFP (10 µg/ml; Molecular Probes A21311), Donkey anti-Rat Alexa Fluor 488 (2 µg/ml; Molecular Probes A21208), Donkey anti-Rat Alexa Fluor 568 (2 µg/ml; Abcam, United Kingdom ab175475) Donkey anti-Rabbit Alexa Fluor 568 (2 µg/ml; Molecular Probes A10042), Donkey anti-Rabbit Alexa Fluor 647 (2 µg/ml; Molecular Probes A31573), Donkey anti-Mouse DyLight 405 (7.5 µg/ml; Jackson ImmunoResearch, West Grove, PA 715-475-150) and Donkey anti-Mouse Alexa Fluor 647 (2 µg/ml; Molecular Probes A31571). Negative controls were processed identically, except that the primary antibodies were replaced with Rat IgG2a (Biolegend, San Diego, CA 400502) and Normal Rabbit IgG (Molecular Probes 10500C). IgG controls were not performed for the differentiation experiments.

## Western blot

Cells were washed once with PBS and then lysed directly on the plate by addition of 1x Laemmli sample buffer (2% (w/v) SDS, 10% (v/v) glycerol, 120 mM Tris-HCl pH 6.8) directly to the culture dish (10 µl/cm$^2$), followed by extensive scraping using a pipette tip. Samples were vortexed, centrifuged briefly to recover recalcitrant fluid, boiled for 5 min at 100°C, sonicated on ice for 10 secs at 20% power using a Branson digital sonifier (Model S-450D), centrifuged for 5 min at 14,000 x g to clear the lysates and then stored at −20°C until use. A Bicinchoninic Acid Protein Assay Kit (Sigma B9643) was used to determine protein concentration and all samples were equalised at 30 µg of protein in 22.5 µl of 1x Laemmli. After addition of 2.5 µl of 1 M DTT containing bromophenol blue, 25 µl of sample was loaded per lane on NuPAGE Novex 4–12% Bis-Tris Protein Gels (Life Technologies; 10 wells: NP0321BOX or 20 wells: WG1402BOX). Electrophoresis was performed in 1x NuPAGE MES buffer without antioxidant at 80 V for 5 min and then 190 V for 45 min. Resolved protein was transferred to Hybond-LFP PVDF membrane (GE Healthcare, United Kingdom) at 360 mA for 85 min on ice in 25 mM Tris base, 190 mM Glycine, 20% Methanol. After three washes in 20 mM Tris (pH 7.5), 150 mM NaCl, 0.1% Tween 20 (TBST), membranes were blocked for >1 hr at RT in TBST containing 10% Skim milk powder. All subsequent antibody incubations were performed in TBST containing 5% BSA and followed by three washes in TBST. H3K27me3 and pan-H3 were probed using Rabbit anti-H3K27me3 (2 µg/ml; Millipore 07–449) or Mouse anti-Histone H3 (0.5 µg/ml; Abcam 10799) and then labelled using Donkey anti-Rabbit Alexa Fluor 488 (0.4 µg/ml; Molecular Probes A21206) or Donkey anti-Mouse Alexa Fluor 647 (0.4 µg/ml; Molecular Probes A31571), respectively. All antibody solutions were centrifuged at >14,000 x g for 15 min prior to use to remove antibody aggregates. Blots were imaged on a Chemidoc MP (Biorad, Hercules, CA), and then quantified using ImageJ. To

ensure comparable densitometry results, gel electrophoresis, protein transfer and antibody staining were performed in parallel for all samples used to generate each Western blot figure panel.

## RNA extraction and microarray processing

RNA was extracted from 0.5–1 $\times$ 10$^6$ sorted ESCs using an RNeasy Mini kit according to manufacturer's instructions (Qiagen). 1 μg of purified RNA was mixed with RNA standards (One Colour RNA Spike-In Kit; Agilent, Santa Clara, CA – 5188–5282) and the mixture labelled with cyanine 3 (Cy3) using the Amino Allyl MessageAmp II with Cy3 kit (Ambion, Austin, TX; AM1795) according to manufacturer's instructions. For each sample, 1.6 μg of labelled RNA was hybridised to a Mouse GE 4x44K v2 microarray (Agilent; G2519F-026655). For short term inhibitor experiments, 150 ng of purified RNA was mixed with RNA standards (Agilent; 5188–5282) and labelled with the LowInput Quick-Amp Labeling Kit One-Color (Agilent; 5190–2305) according to manufacturer's instructions. For each sample, 600 ng of labelled RNA was hybridised to a Mouse GE 8x60k v1 microarray (Agilent; G4852A). RNA quantification and assessment of integrity was performed using the total RNA nano chip (Agilent; 5067–1511) measured on an Agilent Bioanalyzer. Microarrays were hybridised for 17 hr, washed according to manufacturer's instructions and scanned on a Nimblegen scanner at 5 μm resolution (GE 4x44k v2 microarrays) or on a Surescan G2600D scanner using Agilent Scan Control 9.1.7.1 software with default settings (GE 8x60k microarrays). The resulting single channel TIFF images were processed using feature extraction software (Agilent).

All microarray expression data was mapped to Refseq gene identifiers using the biomart database (Ensembl release 65 for NCBI37 [*Kasprzyk, 2011*]) and analysed using the limma package, Bioconductor and custom scripts in R (*Source code 1*) (*Gentleman et al., 2004*; *Ritchie et al., 2015*) (http://www.R-project.org/).

## Differential expression analysis for Hex sorted ESC populations

Published expression data (*Canham et al., 2010*) was obtained from the GEO repository (accession no - GSE13472) using the GEOquery package in R (*Davis and Meltzer 2007*). Differentially expressed genes were identified between the HV$^-$S$^+$ and HV$^+$S$^-$ populations using analysis of variance (ANOVA) with an FDR of 0.05 and a fold change of 1.5. From this initial gene list, we selected only those genes that showed differential expression between HV$^-$S$^+$ and HV$^+$S$^+$ ESC populations that was consistent with the direction of differential expression in the HV$^-$S$^+$ vs. HV$^+$S$^-$ ESC populations across the expression data generated in this study and the published datasets (*Canham et al., 2010*). The complete dataset comprised four biological replicates (independent cultures) composed of two independent Hex reporter clones.

## Quantitative RT-PCR validation of microarrays

First-strand synthesis was performed on 1 μg of total RNA from the samples isolated for the microarray using SuperScript III reverse transcriptase (Life Technologies) according to the manufacturer's instructions. A mix of concentrated cDNA from all samples was used to generate standard curves and cDNA equivalent to 20 ng of total RNA was used in each reaction. Amplification was detected using the Universal Probe Library system on a Roche LightCycler 480 (Roche, Germany). The data was analysed using the methods described in (*Larionov et al., 2005*). See *Supplementary file 5* for a list of primers and probes used.

## Differential expression analysis in Rex sorted ESC populations

Summary gene expression values generated from published RNA-seq data (*Marks et al., 2012*) was kindly provided by H. Marks and H. Stunnenberg. Based on the published analysis, genes with a read depth (Reads Per Kilobase per Million mapped reads - RPKM) less than 0.5 were removed. Differential expression was determined based on the ratio of RPKM values in Rex+ vs. Rex- ESCs using a minimum fold change of 2 as a cut-off.

## Differential expression analysis in ESCs following short term inhibition

RNA was isolated from 3 biological replicates (independent cultures) using an RNeasy Mini kit according to manufacturer's instructions (Qiagen, Germany) and initial cell lysis was achieved by adding buffer RLT (Qiagen) directly to the wells after 24 h of treatment in the indicated conditions and a

single PBS wash. Raw intensity values were extracted from the scanned images using feature extraction software (Agilent Feature Extraction 11.0.1.1) and default parameters (AgilentG3_HiSen_GX_1-color). Intensity values for duplicate probes with the same DNA sequence were replaced with their arithmetic mean. Final processed intensity values were produced using the limma package (R/Bioconductor) by background correction ('normexp' method with an offset of 10) followed by quantile normalisation across all samples. Fold-changes and p-values for differential expression were determined using the empirical Bayes statistics for differential expression (eBayes) framework. The Benjamini-Hochberg method was used to adjust p-values for multiple testing. Differential expression was defined as fold change > 1.5 or < 0.67 and an adjusted p-value of < 0.05. Three biological replicates in the form of three independently grown ESC lines were used per condition.

## Functional gene enrichment analysis

Functional enrichment analysis was performed on lists of differentially expressed genes (gene symbol annotation) using the R interface to the g:Profiler tool (*Tariq et al., 2013*). Analysis was performed against a background of all arrayed genes and with moderate stringency hierarchical filtering. Only those functional terms with a false discover rate (FDR) corrected p value of $\leq 0.05$ were considered to be enriched (g:Profiler applies a Fisher's one-tailed test for assessing statistical significance). Functional enrichment was performed for Biological Processes (BP) and KEGG Pathways.

## Single cell qPCR

Single SSEA1+ cells were sorted into 96 well plates containing 5 µl 2x CellsDirect reaction mix and stored at -80°C until use. After thawing, the plates were processed according to the manufacturer's instructions using 48 DELTAgene assays (*Supplementary file 5*). Each sorted 96 well plate was split in two identical parts to match loading onto 48.48 gene expression Dynamic Arrays (Fluidigm, South San Francisco, CA) and each 48 well section contained 14x single cells and 100 and 1000 cell controls for each cell line. Arrays were primed on an IFC Controller HX (48.48/digital; Fluidigm) and then scanned on a Biomark HD reader (Fluidigm). Fluidigm's Real Time PCR Analysis software was used to assess melting curves and assays with multiple peaks were excluded. The remaining assays were then analyzed in R using the SINGuLAR Analysis Toolset 3.0.3 (Fluidigm). Outlier cells were removed from the analysis using the Outlier Analysis tool from this toolset. Technical information for single cell qPCR assays are outlined in *Supplementary file 5*.

## Imaging and image processing

Immunofluorescence images were taken using a Leica TCS SP8 confocal microscope and identical acquisition settings optimised to minimise out of range pixels were used for all samples. Images were processed/enhanced with rolling ball background subtraction and contrast/brightness enhancement using identical settings for all images in ImageJ (*Schneider et al., 2012*). The quantification of immunofluorescence signal in *Figure 3D* was done using custom scripts in ImageJ (*Source code 2*) fields of view were analysed for each cell line.

## Chimera generation and imaging

To generate E6.5 chimeric embryos, we used HVHC cells (see the methods section) that were cultured in serum+LIF for 24h at which point the medium was changed to serum+LIF+DMSO, N2B27+LIF+DMSO or N2B27+LIF+EPZ (2.5 µM) for a further 24 hr. The cells were sorted to include only Pecam positive ESCs and five cells of varying sizes (2 large, 1 intermediate and 2 small cells per embryo) were selected for injection into each embryo to avoid introducing a bias based on cell cycle and/or cell size. Injected embryos were placed into E0.5 pseudopregnant females that were sacrificed after 6 days to obtain E6.5 embryos by dissection. The complete chimera set comprised embryos generated from one ESC culture for serum+LIF ('control 1'; embryos introduced into 4 pseudopregnant females) and two independent cultures for each of N2B27+LIF+DMSO and N2B27+LIF+EPZ ('control 2' and 'inhibitor' respectively; each replicate was introduced into 3 pseudopregnant females). Stacks of multiple live E6.5 embryos were acquired at a z-step of 2 µm on a Leica TCS SP8 confocal microscope using a 10x objective. The depth of the stacks was adjusted to match the size of the largest embryo and for each slice mCherry and brightfield images were acquired sequentially. To generate chimera images, mCherry stacks were subjected to rolling ball background

subtraction followed by maximum intensity z-projection of all the slices. The resulting mCherry z-projection was then overlayed onto a single brightfield slice that was in focus and a scale bar of 50 μm was added to the overlay image.

Animal work was carried in accordance with European legislation and was authorized by and carried out under Project License 2013-15-2934-00935/JANNI issued by the Danish Regulatory Authority.

## Native chromatin immunoprecipitation

$0.5$–$1.5 \times 10^6$ flow sorted ESCs were centrifuged at 500 x g for 3 min, washed twice in ice-cold PBS and were resuspended in 200 μl of NBA buffer (85 mM NaCl, 5.5% (w/v) Sucrose, 10 mM Tris-HCl pH 7.5, 0.2 mM EDTA, 0.2 mM PMSF, 1 mM DTT, 1x Protease inhibitors (Calbiochem, 539134-1SET)). Cells were lysed by the addition of an equal volume of NBB buffer (NBA + 0.1% (v/v) NP40; SigmaAldrich - I8896) followed by incubation on ice for 3 min. Nuclei were pelleted at 1000 x g for 3 min at 4°C, then washed with NBR buffer (85 mM NaCl, 5.5% (w/v) Sucrose, 10 mM TrisHCl pH 7.5, 3 mM MgCl2, 1.5 mM CaCl2, 0.2 mM PMSF and 1 mM DTT) and pelleted at 2000 x g for 3 min at 4°C. Nuclei were resuspended in NBR supplemented with RNaseA (to 20 μg/ml) at a concentration of $16.7 \times 10^6$ nuclei per ml and incubated at 20°C for 5 min. Chromatin was then fragmented for 10 min at 20°C with the addition of 0.133 U/μl Micrococcal Nuclease (Boehringer units; SigmaAldrich - N3755-500UN; titrated to give predominantly mono-nucleosomes). Micrococcal digestion was stopped with the addition of an equal volume of STOP bufffer (215 mM NaCl, 10 mM TrisHCl pH 8, 20 mM EDTA, 5.5% (w/v) Sucrose, 2% (v/v) TritonX 100, 0.2 mM PMSF, 1 mM DTT, 2x Protease Inhibitors). Digested nuclei were left on ice overnight to release soluble, fragmented chromatin. Chromatin was then centrifuged at 12,000 x g for 10 min at 4°C and the soluble chromatin (supernatant) transferred to a fresh tube. 20% of the released chromatin was retained as an input reference and the remainder incubated for 3 hr at 4°C on a rotating wheel with either anti-H3K4me3 (07–473; Millipore) or anti-H3K27me3 (07–449; Millipore) antibodies pre-coupled to protein A Dynabeads (Life Technologies; 10002D; 1 μg of antibody bound to 10 μl of Dynabead slurry per immunoprecipitation). Immune complexes bound to beads were washed three times with wash buffer 1 (150 mM NaCl, 10 mM TrisHCl pH 8, 2 mM EDTA, 1% (v/v) NP40 and 1% (w/v) sodium deoxycholate) and twice in 1x TE buffer (10 mM Tris pH 8 and 10 mM EDTA) at 4°C on a rotating wheel for 10 min. Chromatin was released from the Dynabeads by incubation with elution buffer (0.1 M NaHCO3 and 1% SDS) for 15 min at 37°C followed by the addition of proteinase K and Tris pH 6.8 (final concentration of 100 μg/ml and 100 mM respectively) and incubation at 55°C overnight. Dynabeads were removed using a magnetic rack and the chromatin was purified using MinElute PCR Purification columns (Qiagen) according to manufacturer's instructions.

## Cross-linked chromatin immunoprecipitation

Approximately $0.5$–$2 \times 10^6$ FACS purified ESCs were centrifuged at 500 x g for 3 min, washed twice with RT PBS and resuspended in 200 μl of culture media. Cells were fixed by the addition of an equal volume of culture media containing 2% methanol free formaldehyde (Thermo Scientific Pierce PN28906; final concentration of 1%) and incubated at RT for 10 min. Fixation was stopped by 5 min incubation with 125 mM glycine at room temperature. Cells were washed in PBS prior to short-term storage at -80°C. All buffers were supplemented with the following just prior to use: 1 mM DTT and 1 x Protease inhibitors (Calbiochem - 539134-1SET). Cell pellets were resuspended in lysis buffer 1 (50 mM Tris-HCl pH 8.1, 10 mM EDTA and 20% SDS) and incubated for 10 min at 4°C. Lysates were diluted 1:10 in ChIP dilution buffer (0.1% Triton X-100, 2mM EDTA, 150 mM NaCl, 20 mM and Tris-HCl pH 8.1) and sonicated with a single 30 s pulse using a soniprep 150 plus (MSE; set to 3.5) followed by a further 7 pulses using a chilled Bioruptor (Diagenode; 1 min cycles of 30 s on / 30 s off on 'high' setting at 4°C). The sonicated extract was pre-cleared by centrifugation at 16,000 x g for 10 min at 4°C. The supernatant was transferred to a fresh tube and supplemented with BSA and triton X-100 to a final concentration of 25 mg/ml and 1% respectively. 10% of the IP volume of the chromatin was retained as an input reference. Anti-H3K27me3 antibody (Millipore 07–449) was pre-coupled to protein A Dynabeads (Life Technologies; 10001D) at a ratio of 1 μg antibody per 10 μl of dynabeads. Antibody-bound beads were added to the fragmented chromatin at 1 μg antibody per million cell equivalents and incubated for 10 hr on a rotating wheel at 4°C. Bead-associated immune

complexes were washed as for native ChIP. Chromatin was released from the beads by incubation with elution buffer (0.1 M NaHCO3 and 1% SDS) for 15 min at 37°C followed by the addition of RNaseA and Tris pH 6.8 (final concentration of 20 mg/ml and 100 mM respectively) and incubation at 65°C for 2 hr followed by the addition of 50 µg of proteinase K and incubation at 65°C for a further 4–6 hr to degrade proteins and reverse the cross-links. Dynabeads were removed using a magnetic rack and the chromatin purified using PCR Purification columns (Qiagen) according to manufacturer's instructions.

## Genome-wide nuclear run-on analysis (GRO-seq)

All buffers were supplemented with RNasin Plus inhibitor to a final concentration of 80 U/ml (Promega, Madison, WI; N2611). Nuclei were prepared from flow sorted ESCs ($3.5 \times 10^6$ per assay) as described for native ChIP (see above). Nuclei were recovered by centrifugation at 1000 x g and incubated in run-on reaction buffer (50 mM Tris-HCl pH 7.5, 10 mM MgCl2, 150 mM NaCl, 25% (v/v) Glycerol, 0.5 mM ATP, 0.5 mM CTP, 0.5 mM GTP, 20 µM BrUTP, 2 mM DTT and 1x Protease inhibitors (Calbiochem, 539134-1SET) for 5 min at RT. In parallel, un-sorted HV cells were incubated in run-on buffer with either BrUTP or UTP (required to control for background RNA signal in the immunoprecipitation). Stop buffer (20 mM Tris pH7.5, 330 mM NaCl, 12 mM EDTA, 1.2% SDS and 40 µg/ml proteinase K) was added to 5x the reaction volume of each sample and incubated for 1 hr at 37°C. Each reaction was sonicated for a single 30 s pulse on high setting using a water bath sonicator at 4°C (Diagenode Bioruptor). RNAs were recovered by acid phenol extraction and isopropanol precipitation followed by DNA digestion (DNA-free kit, Ambion). Br-UTP-incorporated transcripts were immunoprecipitated using a monoclonal anti-BrdU antibody (Sigma B2531) and the isolated RNA reverse transcribed into single stranded DNA using SuperScript III Reverse Transcriptase followed by second strand synthesis using SuperScript Double-Stranded cDNA Synthesis Kit (Life Technologies; 18080044 and 11917–101 respectively).

## Illumina library preparation and sequencing

Native ChIP libraries were prepared as previously described (*Clouaire et al., 2012*) with the following modifications. The end repair step was removed and the A-tailing reaction (reaction 2) was supplemented with 10 U of Polynucleotide Kinase (NEB) and 1 mM ATP. Following heat inactivation, the samples were supplemented with ligation reagents (400 U of T4 DNA ligase (NEB), 1x buffer 2 (NEB), 7.5% (w/v) PEG-6000, 1 mM ATP and 13.3 nM of annealed Illumina adaptors (PE or AU)) and incubated at 16°C overnight. Ligated DNA was purified using MinElute PCR columns (Qiagen) and eluted in $2 \times 10$ µl water. Purified libraries were amplified as previously described (*Clouaire et al., 2012*) and DNA fragments corresponding to mono- to tri-nucleosomes excised from an agarose gel and purified using the QIAEX II gel extraction kit according to manufacturer's instructions (Qiagen).

GRO-seq libraries and cross-linked ChIP-seq libraries were prepared as previously described (*Bowman et al., 2013*) with the following modifications. No purification was performed between the A-tailing reaction and the ligation reactions; instead the ligation was supplemented as described above. Size selection purifications following the ligation and amplification PCR steps were performed with 1x and 0.8x reaction volumes of Agencourt AMPure XP beads (Beckman Coulter - A63880). Multiplexing was performed using multiplex primers 1–6 as described by Bowman et al. (*Bowman et al., 2013*).

ChIP and GRO-seq samples were generated from two replicate ESC cultures (biological replicates) that were cultured and processed independently. Native ChIP replicates were pooled just prior to sequencing whereas cross-linked ChIP and GRO-seq samples were sequenced independently and combined bioinformatically.

H3K4me3 ChIP libraries were captured on an Illumina flow cell for cluster generation and sequenced using an Illumina GAII X Analyzer following the standard Illumina protocol to generate single-end 37 bp reads (performed at the University of Edinburgh Gene Pool). Native H3K27me3 and GRO-seq libraries were sequenced on an Illumina Hi-seq 2000 system following the standard Illumina protocol to generate single-end 50 bp reads (performed at The Danish National High-Throughput DNA Sequencing Centre). Native H3K27me3 sequencing was performed without multiplexing using the primers complementary to the ligated paired-end adaptors. GRO-seq libraries were sequenced as multiplexed samples with primers complementary to the ligated adaptors

(*Bowman et al., 2013*). Cross-linked H3K27me3 libraries were sequenced on an Illumina Hi-seq 4000 system following the standard Illumina protocol to generate single-end 50 bp reads (performed by Beijing Genomics Institute). Cross-linked H3K27me3 ibraries were sequenced as multiplexed samples with primers complementary to the ligated adaptors (*Bowman et al., 2013*).

## Illumina sequence mapping and analysis

### Mapping and normalisation

Single-end sequence reads were mapped to the mouse genome (mm9, NCBI Build 37) using Bowtie2 (*Langmead and Salzberg, 2012*). The resulting files (.sam) were then converted into. bam files using samtools (*Li et al., 2009*) and then into browser viewable. wig files using the 'bedgraph' function in the bedtools package (*Quinlan and Hall, 2010*). Where applicable, replicate datasets were combined into single files. Signal ratios between the IP and matched inputs were calculated for all abutting 50bp windows across the sequenced portion of the genome, median and quantile normalised across ESC populations using the R limma package. Normalised files were used for visualization purposes; all other analyses were performed on raw sequence data and normalised separately to avoid excessive manipulation of the data.

### Heatmap and composite analysis of ChIP-seq and GRO-seq data

Raw sequence density (hits per base) was mapped to Refseq TSSs ($\pm$ 2.5 kb; at 100 bp window resolution) or Refseq gene bodies (gene length $\pm$ 50%; 40 abutting window resolution). Sequence depth was median normalised for each IP/Input pair and converted into $\log_2$ ratios followed by quantile normalisation across samples. Composite profiles for TSSs were generated by plotting the mean $\log_2$(IP/Input) window value for each sample. Heatmaps were generated using a custom R script (*Source code 3*) and ranked based on the magnitude of the modification in the $HV^-S^+$ ESC population. Composite gene body maps depict the mean $\log_2(IP^{HV-S+}/Input^{HV-S+})$-$\log_2(IP^{HV+S+}/Input^{HV+S+})$ for each window.

### Peak finding for H3K27me3

'Peak Finding' for H3K27me3 was performed on ChIP-seq data for both $HV^-S^+$ and $HV^+S^+$ using MACs (version 1.4.1; *Zhang et al., 2008*) with the following parameters: genome size $-2.7 \times 10^9$ bp, tag size $-42$ bp, band-width $-300$ bp, fixed background lambda value and a MACs score cut-off of 1000. Identified sites were merged into a single set of H3K27me3 enriched genomic intervals and mapped to TSSs (Refseq MM9 $\pm$ 100 bp) prior to further analysis.

### Statistical tests

*Kolmogorov–Smirnov (KS)*, Wilcoxon and Fisher's tests were performed in R using base utilities. When comparing the distribution of ChIP-seq and GRO-seq signals in the sorted populations with respect to gene location, statistical significance was tested by randomly permuting the data (10,000 simulations) and applying a K-sample permutation test (oneway_test) in the 'coin' package in R.

### Serum vs. 2i media H3K27me3 analysis

Published H3K27me3 ChIP-seq data (*Marks et al., 2012*; *Kumar et al., 2014*) was downloaded from the short read archive (http://www.ncbi.nlm.nih.gov/sra; accessions - SRR1557682, SRR1557689, SRR064958, SRR064959, SRR064960 and SRR064961). Fastq files were generated for each of these datasets using 'fastq-dump' from the SRA toolkit (Windows version 2.5.2) with the –Z parameter. Single-end sequence reads were mapped to the mouse genome (mm9, NCBI Build 37) using Bowtie2 (*Langmead and Salzberg, 2012*). The resulting files (.sam) were then converted into HOMER tag directories using 'makeTagDirectory' in the Homer toolset (*Heinz et al., 2010*) with –fragLength 150 and the default normalisation behaviour (samples normalised to 10 million mapped reads). The mouse genome was stratified into TSS ($\pm$ 5 kb; CGI or non-CGI [*Illingworth et al., 2010*]), intragenic and intergenic intervals based on Refseq gene annotation. Read coverage for each of these intervals was then determined using 'annotatePeaks' in the Homer toolset with the genome build set to 'none' and –d parameter. Quantification data was plotted in R.

## Data availability

Microarray expression and Illumina sequencing data was deposited in the GEO repository (http://www.ncbi.nlm.nih.gov/geo/) under the accession number: GSE65289.

## Acknowledgements

We would like to thank Elisabeth Freyer and Gelo Dela Cruz for FACS, William Hamilton for EED Western blot control samples and help with cloning and H Marks and H Stunnenberg for sharing their expression data with us. We thank the Danish National High-Throughput DNA Sequencing Centre, SNM, KU for their support with sequencing. This work was supported by a joint grant between JMB and WAB from the BBSRC (BBSRC_BB/H005978/1 (JMB) and BBSRC_BB/H008500/1 (WAB)). WAB is also funded by a unit programme grant from the MRC, UK and JMB by the Novo Nordisk Fonden (NNF), section on basic stem cell research and an NNF project grant 107012.

## Additional information

### Funding

| Funder | Grant reference number | Author |
| --- | --- | --- |
| Biotechnology and Biological Sciences Research Council | BBSRC_BB/H008500/1 | Robert S Illingworth Wendy A Bickmore |
| Medical Research Council | | Robert S Illingworth Wendy A Bickmore |
| Biotechnology and Biological Sciences Research Council | BBSRC_BB/H005978/1 | Jurriaan J Hölzenspies Joshua M Brickman |
| Novo Nordisk Foundation | 107012 | Joshua M Brickman Jurriaan J Hölzenspies |
| Novo Nordisk Foundation | Section for Basic Stem Cell Biology | Jurriaan J Hölzenspies Joshua M Brickman Fabian V Roske |

The funders had no role in study design, data collection and interpretation, or the decision to submit the work for publication.

### Author contributions

RSI, Prepared the ChIP-seq and GRO-seq samples and carried out the bioinformatic analysis, Experimental design and writing the paper, Conception and design, Acquisition of data, Analysis and interpretation of data, Drafting or revising the article; JJH, Performed ChIPs, immunofluorescence, Single cell qRT-PCR analysis and inhibitor Experiments, Experimental design and writing the paper, Conception and design, Acquisition of data, Analysis and interpretation of data, Drafting or revising the article; FVR, Generated reporter ESC lines and CRISPR knockouts, Performed cloning and selection of HVHC, HFHC, HFHCV and HV5.1 Eed mutant ESCs, Experimental design and writing the paper; WAB, JMB, Experimental design and writing the paper, Conception and design, Analysis and interpretation of data, Drafting or revising the article

### Author ORCIDs

Joshua M Brickman, http://orcid.org/0000-0003-1580-7491

### Ethics

Animal experimentation: Animal work was carried in accordance with European legislation and was authorized by and carried out under Project License 2013-15-2934-00935/JANNI issued by the Danish Regulatory Authority.

# Additional files

## Supplementary files

• Supplementary file 1. Genes with differential expression in Epi- and PrEn primed ESCs

• Supplementary file 2. Functional analysis of Epi- and PrEn priming genes

• Supplementary file 3. Genes with differential expression in Rex+ and Rex- cell populations

• Supplementary file 4. Genes showing differential gene expression in ESCs following short-term inhibition with EPZ.

• Supplementary file 5. Quantitative single-cell and qRT PCR assay information.

• Source code 1. Custom R script used to identify genes with differential expression levels between experimental conditions.

• Source code 2. Custom image J script used to extract intensity information from confocal immuno-fluorescence images.

• Source code 3. Custom R script used to generate heatmaps from pre-formatted quantitation data files.

## Major datasets

The following datasets were generated:

| Author(s) | Year | Dataset title | Dataset URL | Database, license, and accessibility information |
|---|---|---|---|---|
| Illingworth RS, Hölzenspies JJ, Roske FV, Bickmore WA, Brickman JM | 2016 | GROseq analysis of nascent transcript abundance in Epi- and PrEn primed mouse ESCs | http://www.ncbi.nlm.nih.gov/geo/query/acc.cgi?acc=GSE65288 | Publicly available at the NCBI Gene Expression Omnibus (GSE65288) |
| Illingworth RS, Hölzenspies JJ, Roske FV, Bickmore WA, Brickman JM | 2016 | Gene expression data for mouse ESCs treated short term (24h) with the EZH2 inhibitor EPZ6438 | http://www.ncbi.nlm.nih.gov/geo/query/acc.cgi?acc=GSE76582 | Publicly available at the NCBI Gene Expression Omnibus (GSE76582) |
| Illingworth RS, Hölzenspies JJ, Roske FV, Bickmore WA, Brickman JM | 2016 | ChIP-seq data for H3K27me3 and H3K4me3 in Epi- and PrEn primed mouse ESCs | http://www.ncbi.nlm.nih.gov/geo/query/acc.cgi?acc=GSE65288 | Publicly available at the NCBI Gene Expression Omnibus (GSE65288) |
| Illingworth RS, Hölzenspies JJ, Roske FV, Bickmore WA, Brickman JM | 2016 | Gene expression data for Epi- and PrEn primed mouse ESCs | http://www.ncbi.nlm.nih.gov/geo/query/acc.cgi?acc=GSE64939 | Publicly available at the NCBI Gene Expression Omnibus (GSE64939) |

The following previously published datasets were used:

| Author(s) | Year | Dataset title | Dataset URL | Database, license, and accessibility information |
|---|---|---|---|---|
| Marks H, Kalkan T, Menafra R, Denissov S, Jones K, Hofemeister H, Nichols J, Kranz A, Stewart AF, Smith A | 2012 | H3K27me3 ChIP-seq data from mouse ESCs and ESC sub-populations cultured in Serum LIF or 2i media | http://www.ncbi.nlm.nih.gov/geo/query/acc.cgi?acc=GSE23943 | Publicly available at the NCBI Gene Expression Omnibus (GSE23943) |

| Kumar RM, Cahan P, Shalek AK, Satija R, DaleyKeyser AJ, Li H, Zhang J, Pardee K, Gennert D, Trombetta JJ | 2014 | H3K27me3 ChIP-seq data from mouse ESCs cultured in Serum LIF or 2i media | http://www.ncbi.nlm.nih.gov/geo/query/acc.cgi?acc=GSE60749 | Publicly available at the NCBI Gene Expression Omnibus (GSE60749) |
| --- | --- | --- | --- | --- |

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
