## [Decision Letter]

Thank you for submitting your article "Polycomb Underlies Transcriptional Heterogeneity in Embryonic Stem Cells" for consideration by *eLife*. Your article has been reviewed by three peer reviewers, and the evaluation has been overseen by a Reviewing Editor and Fiona Watt as the Senior Editor.

The reviewers have discussed the reviews with one another and the Reviewing Editor has drafted this decision to help you prepare a revised submission.

Summary:

In this manuscript, Illingworth and colleagues work towards understanding the transcriptional regulation of cell fate heterogeneity within embryonic stem (ES) cell populations. Specifically, they take advantage of two interconverting populations of mouse ESCs primed for the primitive endoderm or the epiblast lineages, respectively, to gain insights into mechanisms underlying lineage priming in self-renewing ESCs. The authors provide evidence that this heterogeneity correlates with transcriptional variation between the two cell populations, which correlates with subtle changes in the distribution of the H3K27me3 histone post-translational modification (PTM), a mark catalyzed by the Polycomb Repressive Complex 2 (PRC2), both within gene bodies and, for primitive endoderm primed ESCs, at the promoter region. The authors provide evidence that loss of function of Polycombs leads to a loss of transcriptional heterogeneity, and, using both in vitro and in vivo developmental assays, report that transient inhibition of the PRC2 Polycomb complex in the ESC state leads to perturbations in cell fate decisions.

In summary, the authors suggest that polycombs create a block to PrEn commitment. The strength of the work is that the authors reveal a role of polycomb in regulating heterogeneity and lineage priming in self-renewing ESCs. To date most work on polycomb proteins has been done in the context of its function in ESC differentiation, i.e. when ESCs are moved down a specific lineage. As such the manuscript is an important addition for the field and new insights in this area improve our understanding of the early stages of development when pluripotent stem cells undergo lineage commitment.

Essential revisions:

Among the reviewers, there is considerable concern about the low-level effects, which limit the strength of the conclusions that can be reached. While there is not much that the authors can do about this, it limits the ability of the reader to know to what extent the differences are meaningful or direct. So far there are correlations between K27me3 and transcription as well as functional changes in the Polycomb KO, but a clear mechanistic framework is still lacking. Thus, all reviewers agree that more experiments are necessary to support and strengthen the conclusions.

1) The H3K27me3 ChIP data need to be upgraded. First, it is unclear how many replicates were performed to determine the differences in Figure 2. Second, the ChIP-seq of histone PTMs is not a quantitative measure. Instead, ChIP-RX (Orlando et al., Cell Reports, 2014) is the now the accepted technique to establish whether there are quantitative differences in histone PTM levels, in this case for H3K27me3 in the Epi or PrEn cells. In fact, ChIP-RX would be particularly important in this case because the observed modest differences in H3K27me3 levels at promoters and gene bodies between the two cell populations (Figure 2).

2) One question relates to the two populations sorted. How stable are the populations after sorting, what is their transition time? The authors should include further analysis of their sorted cells to show how their changes correlate with the changes in H3K27me3.

3) To better examine the role of Polycomb proteins in maintaining transcriptional heterogeneity, the authors provide single cell qPCRs for some genes. However, it is difficult to appreciate the extent of heterogeneity from the data provided in Figure 3. The minor fluctuations in transcription observed in Figure 3 and Figure 3 do not yet suggest any significant difference (here is no significance testing done for differences)? Could the authors represent their data in Figure 3 in a more quantitative manner? Quantifications of Nanog in several different fields are required in Figure 3 so as to provide an objective quantification of the experiment. Are Rex1 or Hhex now homogeneously expressed? Importantly, the analysis should be carried out in Polycomb knockout cells that have been sorted into either Epi or PrEn lineages, as done in Figure 1. This would require more work, but would be essential to support the statements made by the authors. Perhaps a quick way to do this is to use CRISPR on the already made and validated Hhex reporter ES cell lines to KO Eed and Ring1A, separately.

4) The authors show in Figure 4 that H3K27me3 signal is down by western blot in their inhibition studies, but not what is happening at the level of chromatin. It is impossible to know if the effects of this drug are direct or indirect or, whether cell fate decisions are modified by gene body H3K27me3 status, promoter methylation, or another mechanism. It would be useful to confirm that H3K27me3 is lowered gene bodies/promoters especially in the inhibitor treated cells. More molecular analysis of what is happening at the chromatin level in the inhibitor experiment is important for interpretation of the data.

5) It remains somewhat unclear what the effect of gene body methylation versus promoter methylation is – it would be important for the authors to provide more insights.

[Editors' note: further revisions were requested prior to acceptance, as described below.]

Thank you for resubmitting your work entitled "Polycomb Underlies Transcriptional Heterogeneity in Embryonic Stem Cells" for further consideration at *eLife*. Your revised article has been favorably evaluated by Fiona Watt as the Senior editor, a Reviewing editor, and three reviewers.

The manuscript has been improved but there are two remaining issues that need to be addressed before acceptance, as outlined below:

All reviewers agree that the small changes observed in the manuscript are an inherent problem of their biological system. Therefore, the authors should add a frank comment on the issue of small changes and how that impacts making conclusions in the Discussion. In addition, a title should be chosen that better reflects the paper and is not as general.

Thus, to get the paper accepted really only a change in title and a brief comment on the small changes issues are required.

*Reviewer #1:*

While lots of work was done, three of my points do not seem to have been addressed. These are:

Major Point 1. The title of the manuscript "Polycomb Underlies Transcriptional Heterogeneity in Embryonic Stem Cells" could be misleading for non-experts. It could imply that Polycombs have a directive, instructive or special role in mediating ES cell population heterogeneity. Indeed, as mentioned above, it is unclear what the authors want to inform us about regarding Polycomb function since we already know that their loss leads to deregulation of cellular identity in multiple cell types and organisms. Therefore, a less vague and more specific title reflecting the new findings in the paper would be more appropriate.

Major point 5. Figure 4. I have a general issue with the lack of novelty with Figure 4. As mentioned above, the PRC2 complex has long been established as a regulator of cell fate decisions through dynamic recruitment to, and repression of, alternative lineage genes. For this reason, the observations in Figure 4 that PRC2 catalytic activity is required for normal development both in vivo and in vitro are to me at least, not novel, informative or surprising.

Furthermore, with regard to their new ChIP-RX data in Figure 1, an average or 'meta-plot' of the enrichments across gene bodies would have helped illustrate the profile of the differences more clearly.

I still have outstanding issues with the very minor differences observed between the two populations and the fact that the authors do not inform on the molecular mechanisms that underlie their results.

*Reviewer #2:*

This revision has done a thorough job of responding to my comments and to those of the other reviewers. The strengths of this paper are that it addresses an important question that is tough to get at experimentally, and makes interesting conclusions regarding the role of the PcG system in different precursor populations. This is a fundamental question in developmental biology, so the paper should be of broad interest. The weakness of the paper continues to be the fact that the effects are small, and therefore the conclusions are not as strong as they would be if there were significant effects. This is the biology of the system, however. The strengths significantly outweigh that weakness and I support publication of the revised paper.

*Reviewer #3:*

The authors have done a good job of addressing the reviewers’ concerns although the issue of small changes remain. As pointed out by the authors, this is anticipated and is difficult to address with current technology. I am satisfied that the authors have done everything they can to address this critical issue. It would be good to emphasize the technical issues in relation to the small but reproducible changes in the manuscript. The Discussion section would be a good place to make this addition. I think this will allow the reader to understand the significance of the results better.

---

## [Author Response]

*Essential revisions:*

*Among the reviewers, there is considerable concern about the low-level effects, which limit the strength of the conclusions that can be reached. While there is not much that the authors can do about this, it limits the ability of the reader to know to what extent the differences are meaningful or direct. So far there are correlations between K27me3 and transcription as well as functional changes in the Polycomb KO, but a clear mechanistic framework is still lacking. Thus, all reviewers agree that more experiments are necessary to support and strengthen the conclusions.*

While we agree that some of the effects are small, it is important to keep in mind that lineage priming is by definition a small transcriptional change with functional consequences. It is therefore exceedingly difficult to approach priming on a gene by gene basis using current technology. However, the power of our analysis lies in our ability to define gene sets that vary synchronously with priming and this is how we have been able to substantiate these changes in previous publications (Canham 2010, Morgani 2013). Here we have used the power of these priming gene sets to define two novel modes of Polycomb regulation, a subtle but robust change in the gene body of all priming genes that is inversely correlated with transcription, and a promoter-specific road block to Primitive Endoderm (PrEn) commitment.

However, we appreciate the reviewers concern and have addressed these issues with additional data. In particular, in our response to point 2 raised below we now show that the small differences in gene body H3K27me3 disappear as priming reverses itself during self-renewal, providing further support for the dynamic nature of this Polycomb activity and its coupling to lineage priming.

*1) The H3K27me3 ChIP data need to be upgraded. First, it is unclear how many replicates were performed to determine the differences in Figure 2. Second, the ChIP-seq of histone PTMs is not a quantitative measure. Instead, ChIP-RX (Orlando et al., Cell Reports, 2014) is the now the accepted technique to establish whether there are quantitative differences in histone PTM levels, in this case for H3K27me3 in the Epi or PrEn cells. In fact, ChIP-RX would be particularly important in this case because the observed modest differences in H3K27me3 levels at promoters and gene bodies between the two cell populations (Figure 2).*

The existing ChIP dataset was derived from two independent biological replicates mixed together in equal amounts prior to sequencing. This is now stated in the first paragraph of the subsection “Priming Genes Are Polycomb Targets”.

We agree that, given the subtlety of the described effect, additional replication would strengthen these observations. Consequently, we prepared two additional H3K27me3 ChIP-seq libraries from the primed populations, this time using cross-linked, in place of native, ChIP so that we could spike in pre-prepared cross-linked *Drosophila* material as per the ChIP-RX protocol. The ChIP worked well and corroborated our initial experiments using an equivalent normalisation (new data is now present as “cross linked ChIP,” in Figure 2).

As described, the ChIP-RX protocol is difficult to apply to sorted populations. In the two examples provided by Orlando et al. (Orlando et al. 2014) they use 1.5 and 2.0 x 10^7^ cells for each experiment, and then add 50% of these values in cross-linked S2 cells to provide the *Drosophila* chromatin that enables normalization. These numbers are difficult to obtain in cell sorting experiments. As a result, we tried to scale the ChIP-RX down to cell numbers that could be obtained in multiple cell sorting experiments (between 0.5-1.3 x 10^6^ cells). We also scaled down the number of cross-linked S2 cells (*Drosophila* spike), but as a result, we found that the test to spike ratios varied significantly between samples. Consequently, the resulting normalisation (as outlined by Orlando and colleagues) led to highly variable results, even between replicate experiments. Despite this issue, we could still detect the differential signal between the primed populations, but the overall ChIP enrichment was skewed due to differences in test to spike ratios between samples, making the data harder to interpret. We would be happy to provide spike-normalised data to illustrate the above points, should the reviewers wish to see it or consider it worth publishing in the online response document.

As exogenous spike normalisation is designed to account for global changes in ChIP signals that occur between different cell types, and primed ESCs are essentially the same cell type, we think it is unlikely that it is necessary in this case.

To address the reviewers’ concerns regarding the robustness of the effect we observed, we have provided additional replicates generated using an alternative ChIP methodology that corroborate our results. We hope this is sufficient to satisfy reviewers as to the robust nature of these results. We have also added additional cross-linked ChIP-seq comparing Epiblast (Epi) primed and spontaneously differentiated PrEn (Figure 2—figure supplement 3, new panels C and D).

*2) One question relates to the two populations sorted. How stable are the populations after sorting, what is their transition time? The authors should include further analysis of their sorted cells to show how their changes correlate with the changes in H3K27me3.*

This is an important point, as it is central to understanding the dynamics of lineage priming in ESCs and how this relates to the observed differences in H3K27me3 at gene bodies.

To address this, we utilised two new Hhex reporter lines to monitor the inter-conversion dynamics of the two primed populations. These cell lines contain two different fluorescent proteins (Venus vs. H2B-Cherry) to ensure that our kinetic measurements are independent of the behaviour of different fluorescent proteins. These cell lines also include a constitutively expressed lineage label, that enables us to conduct cell mixing experiments to determine whether the transition between one primed population and another is the same regardless of whether the starting point is a mixed culture or a single population. This has allowed us to exclude the impact of paracrine selective pressure driving the re-establishment of heterogeneity.

Our measurements of the dynamics of priming (Figure 2—figure supplement 4) indicate that sorted ESC populations were largely equivalent after 72 hours of re-culture, although full equilibration requires approximately 5 days (extrapolated from Figure 2—figure supplement 4). Re-equilibration dynamics were equivalent regardless of which population was used as a starting point or whether the opposite populations were mixed together using the different constitutive lineage labels to separate them upon analysis (subsection “Priming Genes Are Polycomb Targets”, last paragraph).

To measure the re-equilibration of gene body H3K27me3, we chose an intermediate time point (30 hours) of re-culture. Thus we took half of the primed populations isolated to address point 1 above, so that the data would be directly comparable, and placed them back in self-renewing culture. Following 30 hours of re-culture, these cells were processed for ChIP-seq (Figure 2—figure supplement 3). Strikingly, the re-equilibration of the primed-populations in this time frame resulted in a complete homogenisation of the differential H3K27me3 levels seen between the primed populations. From this analysis we conclude that: ‘differential gene-body H3K27me3 levels are closely coupled to the dynamics of the cells as they transit between primed states, supporting the idea that these patterns are dictated largely by gene transcription.’

This data is included as a new supplemental figure (Figure 2—figure supplement 4), which is now described in the ‘Priming Genes Are Polycomb Targets’ Results section (last paragraph). We think this constitutes a major improvement to the paper and thank the reviewers for their suggestion.

*3) To better examine the role of Polycomb proteins in maintaining transcriptional heterogeneity, the authors provide single cell qPCRs for some genes. However, it is difficult to appreciate the extent of heterogeneity from the data provided in Figure 3. The minor fluctuations in transcription observed in Figure 3 and Figure 3 do not yet suggest any significant difference (here is no significance testing done for differences)? Could the authors represent their data in Figure 3 in a more quantitative manner?*

We have added density plots representing the per cell expression values of WT, Ring1B KO and Eed KO ESCs (new Figure 3). The single cell data was evaluated for each gene using a Kolmogorov–Smirnov (KS) test under the null hypothesis that there was no shift in the distribution of expression values between the WT and each mutant cell line. This non-parametric test assesses the cumulative distribution of a set of values to determine the likelihood that two samples are derived from the same distribution. Significant comparisons are indicated within the plots (Figure 3). We also compared the p values for the KS tests to those obtained from Wilcoxon-Rank Sum tests (an alternative non-parametric test) performed on the same data. Data from these two statistical tests are found in a new supplemental figure (Figure 3—figure supplement 1).

We have moved the violin plot representation of the single cell qRT-PCR data from Figure 3 into Figure 3—figure supplement 1, to make room for the new analysis in the main figure.

*Quantifications of Nanog in several different fields are required in Figure 3 so as to provide an objective quantification of the experiment.*

This quantification has been inserted as Figure 3 and includes the results of Wilcoxon Rank Sum tests, which were used to test for differential IF signal between the WT and mutant ESCs.

*Are Rex1 or Hhex now homogeneously expressed? Importantly, the analysis should be carried out in Polycomb knockout cells that have been sorted into either Epi or PrEn lineages, as done in Figure 1. This would require more work, but would be essential to support the statements made by the authors. Perhaps a quick way to do this is to use CRISPR on the already made and validated Hhex reporter ES cell lines to KO Eed and Ring1A, separately.*

As the reviewers suggested, we used the CRISPR targeting system to generate new HV5.1 reporter lines lacking either Eed or Ring1B (predominant PRC1 catalytic component in ESCs). We have now included data for an Eed deficient Hhex reporter.

In the time allowed for revision, we have been able to obtain an Eed mutant, but did not obtain a homozygous mutant Ring1B line. We believe that part of the problem may have been a failure of the Ring1B lines to stabilize in ESC culture, but it could also be that we need to go back and try new gRNAs. If this is a sticking point with the reviewers, we can go back and do this inducibly, as we think this might be required to ensure that we obtain an otherwise normal cell line in a relatively short span of time. We believe that this would be beyond the scope of the current manuscript and we felt it best to proceed with the Eed mutant, particularly since it exhibits aspects of the Polycomb phenotype we had already described. We hope the reviewers will be satisfied with the new data we have provided on Eed.

Analysis of the HV reporter Eed mutant is now presented in Figure 3—figure supplement 2 and discussed at the end of the ‘Polycomb is Required for Transcriptional Heterogeneity in Mouse ESCs’ Results section.

Importantly, in Eed mutant reporters, we observe a similar reduction in Hhex heterogeneity as we described in our single cell PCR (Figure 3—figure supplement 2). As a result we do not believe that it would be informative to sort primed populations from these mutants. However, we also observe that the Hhex reporter was significantly induced in Eed mutants in response to adding the Gsk3 inhibitor Chiron (Chi) to media. Chi is known to stabilize β-catenin, leading to the induction of Wnt targets, and Hhex is a known Wnt target (Zorn et al. 1999; Huelsken et al. 2000; Rodriguez et al. 2001; Zamparini et al. 2006). The enhanced induction of the reporter supports one of the substantial conclusions of our paper, that Polycomb acts to suppress endoderm commitment and while PrEn genes are not significantly induced in its absence, they become poised for activation.

*4) The authors show in Figure 4 that H3K27me3 signal is down by western blot in their inhibition studies, but not what is happening at the level of chromatin. It is impossible to know if the effects of this drug are direct or indirect or, whether cell fate decisions are modified by gene body H3K27me3 status, promoter methylation, or another mechanism. It would be useful to confirm that H3K27me3 is lowered gene bodies/promoters especially in the inhibitor treated cells. More molecular analysis of what is happening at the chromatin level in the inhibitor experiment is important for interpretation of the data.*

To address this point we performed native H3K27me3 ChIP followed by quantitative PCR in 3 independent Hhex reporter lines treated for 24h with either EPZ or DMSO (control). This candidate analysis confirmed the extensive reduction in H3K27me3 levels observed by western blot (Figure 4) and extended this to show that this occurs to an equivalent extent at both TSSs and gene bodies. This new information can be found in the ‘Short-term Inhibition of PRC2 function results in enhanced PrEn priming’ Results section and Figure 4—figure supplement 1 (new panel).

*5) It remains somewhat unclear what the effect of gene body methylation versus promoter methylation is – it would be important for the authors to provide more insights.*

In the manuscript we maintain that gene body H3K27me3 varies during priming inversely with transcription. When PolII density is decreased, H3K27me3 accumulates in the gene body. In the promoter, the unique H3K27me3 configuration around the start site of transcription in the PrEn priming genes enables transcriptional variation, but constrains it, to block commitment. During priming we observed no change in promoter localized H3K27me3, supporting this idea. However, what about differentiation?

To address this we collected the spontaneously differentiating fraction of PrEn from ESCs by isolating the HV^+^S^-^ fraction by FACS and performed H3K27me3 ChIP on these cells. We then compared this data to that of the Epi-primed fraction (HV^-^S^+^). This supports our observations based on the published Rex reporter cells (Figure 2—figure supplement 3) showing more pronounced changes in H3K27me3 levels including changes at the TSS and the region just upstream of it. Based on these results we suggest that ‘changes in the levels of gene body H3K27me3 are inversely proportional to transcriptional changes in both primed and spontaneously differentiating ESCs, while measurable losses at TSSs only become evident upon differentiation’. This new data is presented in Figure 2—figure supplement 3 and discussed in the ‘Priming Genes Are Polycomb Targets’ Results section.

[Editors' note: further revisions were requested prior to acceptance, as described below.]

*All reviewers agree that the small changes observed in the manuscript are an inherent problem of their biological system. Therefore, the authors should add a frank comment on the issue of small changes and how that impacts making conclusions in the Discussion. In addition, a title should be chosen that better reflects the paper and is not as general.*

*Thus, to get the paper accepted really only a change in title and a brief comment on the small changes issues are required.*

We have changed the title to:

"Polycomb Enables Primitive Endoderm Lineage Priming in Embryonic Stem Cells."

We have added a new paragraph to the Discussion:

“Lineage primed cells, although functionally distinct, differ only subtly at the level of gene expression. […] While the polycomb system has long been associated with regulating developmental gene expression, here we show that these proteins are both enablers and regulators of priming toward specific lineages during sequential rounds of ESC self-renewal. ”

*Reviewer #1:*

*While lots of work was done, three of my points do not seem to have been addressed. These are:*

*Major Point 1. The title of the manuscript "Polycomb Underlies Transcriptional Heterogeneity in Embryonic Stem Cells" could be misleading for non-experts. It could imply that Polycombs have a directive, instructive or special role in mediating ES cell population heterogeneity. Indeed, as mentioned above, it is unclear what the authors want to inform us about regarding Polycomb function since we already know that their loss leads to deregulation of cellular identity in multiple cell types and organisms. Therefore, a less vague and more specific title reflecting the new findings in the paper would be more appropriate.*

As mentioned above, we have changed the title.

*Major point 5. Figure 4 have a general issue with the lack of novelty with Figure 4. As mentioned above, the PRC2 complex has long been established as a regulator of cell fate decisions through dynamic recruitment to, and repression of, alternative lineage genes. For this reason, the observations in Figure 4 that PRC2 catalytic activity is required for normal development both in vivo and in vitro are to me at least, not novel, informative or surprising.*

We completely disagree with the reviewer. Figure 4 focuses on the role of PRC proteins in lineage priming. We are not aware of any other study that has addressed this issue, particularly at the level of chimera analysis.

In particular we show that removal of the PRC mark does not have a significant impact on gene expression during self-renewal (Figure 2—figure supplement 2), but impacts the probability cells will differentiate to neural or endodermal cell types. The implication is that loss of PRC2 activity does not drive priming, but enables primed cells to escape into differentiation, when confronted with a signal. We believe that this represents a new way of thinking about polycomb’s role in development, it can be seen as preserving potency at the early stages of differentiation.

*Furthermore, with regard to their new ChIP-RX data in Figure 1, an average or 'meta-plot' of the enrichments across gene bodies would have helped illustrate the profile of the differences more clearly.*

As the levels of H3K27m3 vary significantly across the dataset, we believe the box blots are still the best way to display this difference.

*I still have outstanding issues with the very minor differences observed between the two populations and the fact that the authors do not inform on the molecular mechanisms that underlie their results.*

We have added a paragraph to the Discussion as requested by the editor.